# Embedding Safety into RL: A New Take on Trust Region Methods

## Abstract

Reinforcement Learning (RL) agents are capable of solving a wide variety of tasks, but are prone to produce unsafe behaviour. Constrained Markov Decision Processes (CMDPs) are a popular framework for incorporating safety constraints. However, common solution methods often compromise reward maximization by being overly conservative or by allowing unsafe behaviour during training. We propose Constrained Trust Region Policy Optimization (C-TRPO), a novel approach that modifies the geometry of the policy space based on the safety constraints, yielding trust regions composed exclusively of safe policies and ensuring constraint satisfaction throughout training. We theoretically study the convergence and update properties of C-TRPO and highlight connections to TRPO, Natural Policy Gradient (NPG), and Constrained Policy Optimization (CPO). We demonstrate experimentally that C-TRPO significantly reduces constraint violations while achieving competitive return compared to state-of-the-art algorithms.

## 1 Introduction

Reinforcement Learning (RL) has emerged as a highly successful paradigm in machine learning for solving sequential decision and control problems, with policy gradient (PG) algorithms as a popular approach (Williams, 1992; Sutton et al., 1999; Konda & Tsitsiklis, 1999). Policy gradients are especially appealing for high-dimensional continuous control because they can be easily extended to function approximation. Due to their flexibility and generality, there has been significant progress in enhancing PGs to work robustly with deep neural network-based approaches. Variants of natural policy gradient methods such as Trust Region Policy Optimization (TRPO) and Proximal Policy Optimization (PPO) are among the most widely used general-purpose reinforcement learning algorithms (Schulman et al., 2017a;b).

While flexibility makes PGs popular among practitioners, it comes at a cost: the policy can explore any behavior during training, posing significant risks in real-world applications. Many methods have been proposed to improve the safety of policy gradients, often based on the Constrained Markov Decision Process (CMDP) framework. However, existing methods either struggle to minimize constraint violations during training or severely limit the agent's performance.

This work introduces a simple strategy to enhance constraint satisfaction in trust-region-based safe policy gradient approaches without compromising performance. We propose a novel family of policy divergences, inspired by barrier function methods in optimization and safe control, that modify the policy geometry to ensure that the trust regions consist only of safe policies.

This approach is motivated by the observation that TRPO and related methods base their trust region on the state-average Kullback-Leibler (KL) divergence. It can be derived as the Bregman divergence induced by the negative conditional entropy on the space of state-action occupancies, as shown by Neu et al. (2017). The main insight of the present work is that safer trust regions can be derived by altering this function to incorporate the cost constraints. The resulting divergence is skewed away from the constraint surface, which is achieved by augmenting the negative conditional entropy by another convex barrier-like function. Manipulating the policy divergence in this way allows us to obtain a *provably* safe trust region-based policy optimization algorithm that retains most of TRPO's mechanisms and guarantees, simplifying existing methods, while achieving competitive returns with less constraint violations throughout training.

**Related work**   Classic solution methods for CMDPs rely on linear programming techniques, see Altman (1999). However, they struggle to scale, making them unsuitable for high-dimensional or continuous control problems. While there are numerous approaches to CMDPs, we focus on model-free, direct policy optimization methods. Model-based approaches, like those popularized by Berkenkamp et al. (2017), are attractive due to their strict safety guarantees, but require the learning of a model, which is not always feasible.

*Lagrangian methods* are a widely adopted approach, where the optimization problem is reformulated as a weighted objective that balances rewards and penalties for constraint violations. This is often motivated by Lagrangian duality, where the penalty coefficient is interpreted as the dual variable. Learning the coefficient with stochastic gradient descent presents a popular baseline (Achiam et al., 2017; Ray et al., 2019; Chow et al., 2019; Stooke et al., 2020). However, a naively tuned Lagrange multiplier may not work well in practice due to oscillations and overshoot. To address this issue, Stooke et al. (2020) apply PID control to tune the dual variable during training, which achieves less oscillations around the constraint and faster convergence to a feasible policy. While Lagrangian approaches are becoming increasingly popular, it is not entirely clear how to update the dual variables during training, see Sohrabi et al. (2024).

*Penalty methods* such as IPO (Liu et al., 2020) and P3O (Zhang et al., 2022) propose weighted penalty-based policy optimization objectives based on practical considerations. The penalties are weighted against the reward objective where the penalty coefficient is a hyper-parameter. This simplifies the Lagrangian approach since the penalty coefficients don't have to be optimized during training, which results in improved stability. More recently, the approach to use (smoothed) log-barriers (Usmanova et al., 2024; Zhang et al., 2024a; Dey et al., 2024) became more popular, which keeps the algorithm simple due to the penalty approach, but can offer certain constraint satisfaction guarantees, see e.g. Ni & Kamgarpour (2024). However, working with an explicit penalty produces suboptimal policies w.r.t the original constrained MDP and thus introduces an additional error, which has to be controlled; see for example Geist et al. (2019); Müller & Cayci (2024) for treatments of the regularization error in the unconstrained case, and Liu et al. (2020) for an example of an optimization gap in safe policy optimization. In contrast, changing the trust regions and therefore the problem geometry does not change the objective function and the set of optimizers and therefore does not introduce an additional error.

*Trust region methods* are closely related to our approach, particularly Constrained Policy Optimization (CPO; Achiam et al. (2017)), which extends TRPO by restricting updates to the intersection of the trust region and the safe policy set, which ensures safety during training. While CPO provides constraint satisfaction guarantees, it tends to oscillate around the constraint boundary with high overshoot as it only prevents the policy updates of TRPO from leaving the safe policy set. To address constraint satisfaction, Projection-based CPO (PCPO; Yang et al. (2020)) projects updates onto the safe policy space between updates, improving stability but further hindering reward maximization. Building on PCPO, Zhang et al. (2020) replace second-order updates with a computationally efficient first-order approach, and Yang et al. (2022) further refine these methods with a different projection approach, which achieves improved performance bounds by incorporating Generalized Advantage Estimation (GAE; Schulman et al. (2018)).

**Rethinking safe trust region methods**   We adopt a trust region approach that constructs trust regions exclusively within the safe policy set, eliminating the need for projections or constrained optimization in the inner loop. Trust region methods retain TRPO's update guarantees for both reward and constraints but often underperform compared to barrier penalty methods. To address this, we replace the state-average KL-divergence with policy divergences that act as barrier functions, see Figure 1. This modification encourages updates of the resulting trust region method to move more parallel to the constraint surfaces rather than directly toward and thereby improves constraint satisfaction, simplifies optimization, and achieves competitive returns by maintaining policies within the safe set for longer, see also Figure 6 in the Appendix.

**Contributions**   We summarize our contributions as follows:

- In Section 3, we introduce a modified policy divergence such that every trust region consists of only safe policies. We introduce an idealized TRPO update based on the modified

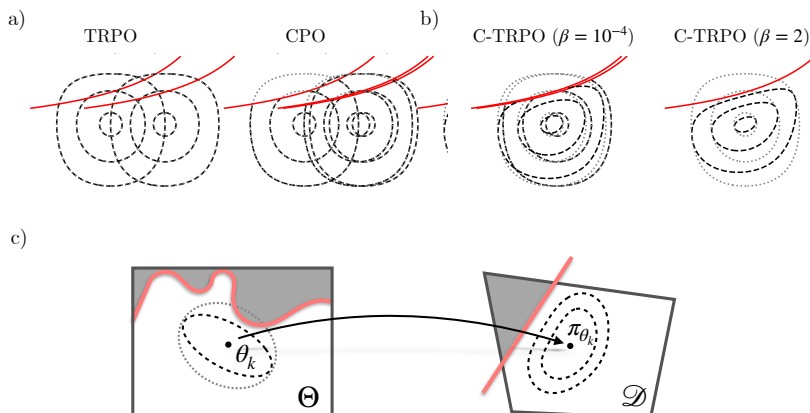

Figure 1: Illustration of policy divergences (dashed) close to the constraint (red). a) TRPO (dotted for reference) and CPO. b) C-TRPO's divergence depends on the hyper-parameter $\beta$, which modulates the strength of the barrier towards the constraint surface. For $\beta \searrow 0$ we obtain an update equivalent to CPO, and more conservative updates for larger values ($\beta = 2$). The plots were generated with the toy MDP in Figure 2. c) Shown are the quadratic approximations of the divergence in parameter space, which is obtained by mapping the policy onto its occupancy measure, where a safe geometry can be defined using standard tools from convex optimization (safe region in white).

> divergence, a tractable optimization algorithm for deep function approximation (C-TRPO), and a corresponding natural gradient method (C-NPG).

- We provide an efficient implementation of the proposed approximate C-TRPO method, see Section 3.2, which comes with a minimal overhead compared to TRPO (up to the estimation of the expected cost) and no overhead compared to CPO. We demonstrate experimentally that C-TRPO yields competitive returns with smaller constraint violations compared to common safe policy optimization algorithms, see Section 5.

- In Section 4, we introduce C-TRPO's improvement guarantees and contrast to TRPO and CPO. Further, we show that the C-NPG method is the continuous time limit of C-TRPO and provides global convergence guarantees towards the optimal safe policy; this is in contrast to penalization or barrier methods, which introduce a bias

## 2 BACKGROUND

We consider the infinite-horizon discounted constrained Markov decision process (CMDP) and refer the reader to Altman (1999) for a general treatment. The CMDP is given by the tuple $(\mathcal{S}, \mathcal{A}, P, r, \mu, \gamma, \mathcal{C})$, where $\mathcal{S}$ and $\mathcal{A}$ are the finite state-space and action-space respectively and we refer to Appendix B.3 for a discussion of continuous state and action spaces. Further, $P \colon \mathcal{S} \times \mathcal{A} \to \Delta_{\mathcal{S}}$ is the transition kernel, $r \colon \mathcal{S} \times \mathcal{A} \to \mathbb{R}$ is the reward function, $\mu \in \Delta_{\mathcal{S}}$ is the initial state distribution at time $t = 0$, and $\gamma \in [0, 1)$ is the discount factor. The space $\Delta_{\mathcal{S}}$ is the set of categorical distributions over $\mathcal{S}$. Further, define the constraint set $\mathcal{C} = \{(c_i, b_i)\}_{i=1}^m$, where $c_i \colon \mathcal{S} \times \mathcal{A} \to \mathbb{R}$ are the cost functions and $b_i \in \mathbb{R}$ are the cost thresholds.

An agent interacts with the CMDP by selecting a policy $\pi \in \Pi$ from the set of all Markov policies, i.e. an element from the Cartesian product of $|\mathcal{S}|$ probability simplices on $\mathcal{A}$. Given such a policy $\pi$, the value functions $V_r^\pi, V_{c_i}^\pi \colon \mathcal{S} \to \mathbb{R}$, action-value functions $Q_r^\pi, Q_{c_i}^\pi \colon \mathcal{S} \times \mathcal{A} \to \mathbb{R}$, and advantage functions $A_r^\pi, A_c^\pi \colon \mathcal{S} \times \mathcal{A} \to \mathbb{R}$ associated with the reward $r$ and the $i$-th cost $c_i$ are defined as

$$V_f^\pi(s) \coloneqq (1 - \gamma) \mathbb{E}_\pi \left[ \sum_{t=0}^\infty \gamma^t f(s_t, a_t) \Big| s_0 = s \right],$$

where the function $f$ is either $r$ or $c_i$, and the expectations are taken over trajectories of the Markov process, meaning with respect to the initial distribution $s_0 \sim \mu$, the policy $a_t \sim \pi(\cdot|s_t)$ and the state

transition $s_{t+1} \sim P(\cdot|s_t, a_t)$. Analogously, we set

$$Q_f^\pi(s, a) := (1 - \gamma) \mathbb{E}_\pi \left[ \sum_{t=0}^\infty \gamma^t f(s_t, a_t) \Big| s_0 = s, a_0 = a \right] \text{ and } A_f^\pi(s, a) := Q_f^\pi(s, a) - V_f^\pi(s).$$

The goal is to solve the following constrained optimization problem

$$\text{maximize}_{\pi \in \Pi} \ V_r^\pi(\mu) \quad \text{subject to} \quad V_{c_i}^\pi(\mu) \le b_i \quad \text{for all } i = 1, \dots, m, \tag{1}$$

where $V_f^\pi(\mu)$ are the expected values under the initial state distribution $V_f^\pi(\mu) := \mathbb{E}_{s \sim \mu}[V_f^\pi(s)]$. We will also write $V_f^\pi = V_f^\pi(\mu)$, and omit the explicit dependence on $\mu$ for convenience, and we write $V_f(\pi)$ when we want to emphasize its dependence on $\pi$. We denote the set of safe policies by $\Pi_{\text{safe}} = \bigcap_{i=1}^m \{\pi : V_{c_i}(\pi) \le b_i\}$ and always assume that it is nontrivial.

**The Dual Linear Program for CMDPs** Any stationary policy $\pi$ induces a discounted state-action (occupancy) measure $d_\pi \in \Delta_{\mathcal{S} \times \mathcal{A}}$, indicating the relative frequencies of visiting a state-action pair, discounted by how far the event lies in the future. This probability measure is defined as

$$d_\pi(s, a) := (1 - \gamma) \sum_{t=0}^\infty \gamma^t \mathbb{P}_\pi(s_t = s) \pi(a|s), \tag{2}$$

where $\mathbb{P}_\pi(s_t = s)$ is the probability of observing the environment in state $s$ at time $t$ given the agent follows policy $\pi$. For finite MDPs, it is well-known that maximizing the expected discounted return can be expressed as the linear program

$$\text{maximize}_d \ r^\top d \quad \text{subject to} \ d \in \mathscr{D},$$

where $\mathscr{D}$ is the set of feasible state-action measures, which form a polytope (Kallenberg, 1994). Analogously to an MDP, the discounted cost CMDP can be expressed as the linear program

$$\text{maximize}_d \ r^\top d \quad \text{subject to} \ d \in \mathscr{D}_{\text{safe}}, \tag{3}$$

where $\mathscr{D}_{\text{safe}} = \bigcap_{i=1}^m \{d : c_i^\top d \le b_i\} \cap \mathscr{D}$ is the safe occupancy set, see Figure 4 in Appendix A.

**Information Geometry of Policy Optimization** Among the most successful policy optimization schemes are natural policy gradient (NPG) methods or variants thereof, such as trust-region and proximal policy optimization (TRPO and PPO, respectively). These methods assume a convex geometry and corresponding Bregman divergences in the state-action polytope, see Neu et al. (2017); Müller & Montúfar (2023) for more detailed discussions. A general trust region update is defined as

$$\pi_{k+1} \in \arg\max_{\pi \in \Pi} \mathbb{A}_r^{\pi_k}(\pi) \quad \text{sbj. to } D_\Phi(d_{\pi_k}||d_\pi) \le \delta, \tag{4}$$

where $D_\Phi : \mathscr{D} \times \mathscr{D} \to \mathbb{R}$ is the Bregman divergence induced by a convex $\Phi : \text{int}(\mathscr{D}) \to \mathbb{R}$, and

$$\mathbb{A}_r^{\pi_k}(\pi) = \mathbb{E}_{s, a \sim d_{\pi_k}} \left[ \frac{\pi(a|s)}{\pi_k(a|s)} A_r^{\pi_k}(s, a) \right], \tag{5}$$

is called the *policy advantage* or *surrogate advantage*. We can interpret $\mathbb{A}$ as a surrogate optimization objective for the expected return. In particular, for a parameterized policy $\pi_\theta$, it holds that $\nabla_\theta \mathbb{A}_{r, \pi_{\theta_k}}(\pi_\theta)|_{\theta = \theta_k} = \nabla_\theta V_r(\theta_k)$, see Kakade & Langford (2002); Schulman et al. (2017a).

TRPO and the original NPG assume the same policy geometry (Kakade, 2001; Schulman et al., 2017a), since they employ an identical Bregman divergence

$$D_K(d_{\pi_1}||d_{\pi_2}) := \sum_{s, a} d_{\pi_1}(s, a) \log \frac{\pi_1(a|s)}{\pi_2(a|s)} = \sum_s d_{\pi_1}(s) D_{KL}(\pi_1(\cdot|s)||\pi_2(\cdot|s)).$$

We refer to Appendix A for details on Bregman divergences. We call $D_K$ the *Kakade divergence* and informally write $D_K(\pi_1, \pi_2) := D_K(d_{\pi_1}, d_{\pi_2})$. This divergence can be shown to be the Bregman divergence induced by the negative conditional entropy

$$\Phi_K(d_\pi) := \sum_{s, a} d_\pi(s, a) \log \pi(a|s), \tag{6}$$

see Neu et al. (2017). It is well known that with a parameterized policy $\pi_\theta$, a linear approximation of $\mathbb{A}$ and a quadratic approximation of the Bregman divergence $D_K$ at $\theta_k$, one obtains the *natural policy gradient* step given by

$$\theta_{k+1} = \theta_k + \epsilon_k G_K(\theta_k)^+ \nabla_\theta V_r(\pi_{\theta_k}), \tag{7}$$

where $G_K(\theta)^+$ denotes a pseudo-inverse of the generalized Fisher-information matrix of the policy with entries given by $G_K(\theta)_{ij} = \partial_{\theta_i} d_\theta \nabla^2 \Phi_K(d_\theta) \partial_{\theta_j} d_\theta$, see Schulman et al. (2017a); Müller & Montúfar (2023) and Appendix A for more detailed discussions.

## 3   A SAFE GEOMETRY FOR CONSTRAINED MDPs

To prevent the policy iterates from violating the constraints during optimization, we construct policy divergences for which the trust regions are contained in the safe policy set.

### 3.1   SAFE TRUST REGIONS

A Bregman divergence is induced by a mirror function that dictates the behavior of the divergence, see A. Take for example the mirror function for TRPO and NPG in Equation (6). The divergence is defined when both policies are in the interior of $\mathscr{D}$, and as either one of the policies approaches the boundary of the state-action polytope, the divergence approaches infinity. Hence, TRPO and NPG don't allow their policy iterates to become entirely deterministic during optimization.   Since the behavior of a Bregman divergences is dictated by the shape of its mirror function, we first construct a family of *safe mirror functions*, that induce policy divergences that are finite only in the safe occupancy set $\mathscr{D}_{\text{safe}}$ instead of the entire state-action polytope $\mathscr{D}$. Safe policy divergences, in turn, let us derive safe trust region and natural policy gradient methods.

To this end, we consider mirror functions of the form

$$\Phi_C(d) := \Phi_K(d) + \sum_{i=1}^m \beta_i \phi(b_i - c_i^\top d), \tag{8}$$

where $\Phi_K$ is the conditional entropy defined in Equation (6), and $\phi \colon \mathbb{R}_{>0} \to \mathbb{R}$ is a convex function with $\phi'(x) \to +\infty$ for $x \searrow 0$. This ensures that $\Phi_C \colon \text{int}(\mathscr{D}_{\text{safe}}) \to \mathbb{R}$ is strictly convex and has infinite curvature at the cost surface $b_i - c_i^\top d = 0$, which means $\|\nabla \Phi_C(d_k)\| \to +\infty$, when $b_i - c_i^\top d_k \searrow 0$. Possible candidates for $\phi$ are $\phi(x) = -\log(x)$ and $\phi(x) = x\log(x)$ corresponding to a logarithmic barrier and entropy, respectively.

The Bregman divergence induced by $\Phi_C$ is given by

$$D_C(d_1 \| d_2) = D_K(d_1 \| d_2) + \sum_{i=1}^m \beta_i D_{\phi_i}(d_1 \| d_2), \tag{9}$$

where

$$D_{\phi_i}(d_1 \| d_2) = \phi(b_i - V_{c_i}(\pi_1)) - \phi(b_i - V_{c_i}(\pi_2)) + \phi'(b_i - V_{c_i}(\pi_2))(V_{c_i}(\pi_1) - V_{c_i}(\pi_2)). \tag{10}$$

The corresponding trust-region scheme is given by

$$\pi_{k+1} \in \underset{\pi \in \Pi}{\arg\max}\, \mathbb{A}_r^{\pi_k}(\pi) \quad \text{sbj. to } D_C(d_{\pi_k} \| d_\pi) \le \delta, \tag{11}$$

where $\mathbb{A}_r$ is defined in Equation (5). Note the constraint is only satisfied if $d_1, d_2 \in \text{int}(\mathscr{D}_{\text{safe}})$ and the divergence approaches $+\infty$ as $d_2$ approaches the boundary of the safe set. Thus, the trust region $\{d \in \mathscr{D} : D_C(d_k \| d) \le \delta\}$ is contained in the set of safe occupancy measures for any finite $\delta$. Analogously to the case of unconstrained TRPO the corresponding natural policy gradient scheme is given by

$$\theta_{k+1} = \theta_k + \epsilon_k G_C(\theta_k)^+ \nabla V_r(\theta_k), \tag{12}$$

where $G_C(\theta)^+$ denotes an arbitrary pseudo-inverse of $G_C(\theta)_{ij} = \partial_{\theta_i} d_\theta^\top \nabla^2 \Phi_C(d_\theta) \partial_{\theta_j} d_\theta$.

## 3.2 Constrained Trust Region Policy Optimization

If we could solve the optimization problem in Equation (11) exactly, we would obtain a provably safe trust region policy optimization method with zero constraint violations, as long as we start with a safe policy. However, the exact trust region update Equation (11) cannot be computed. Firstly, the divergence depends on expected cost values, which we can only estimate. The resulting estimation errors of the divergence might cause the policy iterates to leave the safe set, in which case the divergence becomes ill-defined. Finally, the divergence also depends on the expected cost value of the proposal policy, which is not available during the updates. To address these issues, we propose an update based on a *surrogate divergence*, similar to how surrogate objectives are used in policy optimization. We propose the following update, which we call *Constrained TRPO* (C-TRPO).

$$\pi_{k+1} = \arg\max_{\pi \in \Pi} \mathbb{A}_r^{\pi_k}(\pi) \quad \text{sbj. to } \bar{D}_C(\pi||\pi_k) \leq \delta. \tag{13}$$

Here, $\bar{D}_C$ is the surrogate divergence, defined below. Algorithm 1 shows the implementation of C-TRPO, which performs a constrained trust region update if the current policy is safe or a recovery step that minimizes the cost if the policy is unsafe. For the trust region update, we follow a similar implementation to the original TRPO, estimating the divergence, using a linear approximation of the surrogate objective, and a quadratic approximation of the trust region.

---

**Algorithm 1** Constrained TRPO (C-TRPO); differences from TRPO in blue

1: **Input:** Initial policy $\pi_0 \in \Pi_\theta$, safety parameter $\beta > 0$, recovery parameter $0 < b_H \leq b$
2: **for** $k = 0, 1, 2, \ldots$ **do**
3:      Sample a set of trajectories following $\pi_k = \pi_{\theta_k}$
4:      **if** $\pi_k \in \Pi_{\text{safe}}^H$ **then**
5:          $A \leftarrow A_r$; $D \leftarrow \bar{D}_C = \bar{D}_{\text{KL}} + \beta\bar{D}_\Phi$ {Constrained trust region update}
6:      **else**
7:          $A \leftarrow -A_c$; $D \leftarrow \bar{D}_{\text{KL}}$ {Recovery}
8:      **end if**
9:      Compute $\pi_{k+1}$ using TRPO with $A$ as advantage estimate and with $D$ as policy divergence.
10: **end for**

---

**Surrogate Divergence** To aid in clarity, we focus on the case with a single constraint, but the results are easily extended to multiple constraints by summation of the individual constraint terms. In practice, the exact constrained KL-Divergence $D_C$ cannot be evaluated, because it depends on the cost-return of the optimized policy $V_c(\pi)$. However, we can approximate it locally around the policy of the $k$-th iteration, $\pi_k$, using a surrogate divergence. This surrogate can be expressed as a function of the policy cost advantage

$$\mathbb{A}_c^{\pi_k}(\pi) = \mathbb{E}_{d_{\pi_k}}\left[\frac{\pi(a|s)}{\pi_k(a|s)}A_c^{\pi_k}(s,a)\right], \tag{14}$$

which approximates $V_c(\pi) - V_c^{\pi_k}$ up to first order in the policy parameters (Kakade & Langford, 2002; Schulman et al., 2017a; Achiam et al., 2017). Assume $\pi_k \in \Pi_{\text{safe}}$ and define the *constraint margin* $\delta_b = b - V_c^{\pi_k}$, which is positive if $\pi_k \in \Pi_{\text{SAFE}}$. Further, define the surrogate divergence $\bar{D}_C(\pi||\pi_k) = \bar{D}_{\text{KL}}(\pi||\pi_k) + \beta\bar{D}_\phi(\pi||\pi_k)$, where

$$\bar{D}_{\text{KL}}(\pi||\pi_k) = \sum_{s \in \mathcal{S}} d_{\pi_k}(s)D_{\text{KL}}(\pi||\pi_k) \tag{15}$$

and

$$\bar{D}_\phi(\pi_\theta||\pi_{\theta_k}) = \begin{cases} \phi(\delta_b - \mathbb{A}_c^{\pi_k}(\pi)) - \phi(\delta_b) + \phi'(\delta_b)\mathbb{A}_c^{\pi_k}(\pi), & \text{if } \delta_b - \mathbb{A}_c^{\pi_k} \in \text{dom}(\phi) \\ \infty & \text{otherwise .} \end{cases} \tag{16}$$

The surrogate $\bar{D}_\phi$ is closely related to the Bregman divergence $D_\phi$. They are equivalent up to the substitution $V_c(\pi) - V_c(\pi_k) \to \mathbb{A}_c^{\pi_k}(\pi)$, see Appendix B.1. The surrogate can be estimated from samples of the MDP. In the practical implementation, we estimate $\delta_b$, and the policy cost advantage from trajectory samples using GAE-$\lambda$ estimates Schulman et al. (2018). The consequences of the substitution in the surrogate will be discussed in Section 4.

**Recovery with Hysteresis**   The iterate may still leave the safe policy set $\Pi_{\text{safe}}$, either due to approximation errors of the divergence, or because we started outside the safe set. In this case, we perform a recovery step, where we only minimize the cost with TRPO as by Achiam et al. (2017). In tasks where the policy starts in the unsafe set, C-TRPO can get stuck at the cost surface. This is easily mitigated by including a hysteresis condition for returning to the safe set. If $\pi_{k-1}$ is the previous policy, then $\pi_k \in \Pi_{\text{safe}}^{\text{H}}$ with $\Pi_{\text{safe}}^{\text{H}} = \{\pi_\theta \in \Pi_\theta \text{ and } V_c(\pi_\theta) \leq b_{\text{H}}\}$ where $b_{\text{H}} = b$ if $\pi_{k-1} \in \Pi_{\text{safe}}^{\text{H}}$ and a user-specified fraction of $b$ otherwise.

**Computational Complexity**   The C-TRPO implementation adds no computational overhead compared to CPO, since $\bar{D}_\phi$ is just a function of the cost advantage estimate, and is simply added to the divergence of TRPO. Compared to TRPO, the cost value function must be approximated.

## 3.3 Constrained Natural Policy Gradient

Practically, the C-TRPO optimization problem in Equation (13) is solved like traditional TRPO: the objective is approximated linearly, and the constraint is approximated quadratically in the policy parameters using automatic differentiation and the conjugate gradients algorithm. This leads to the policy parameter update

$$\theta_{k+1} = \theta_k + \alpha^i \sqrt{\frac{2\delta}{g_k^\top H_k^{-1} g_k}} \cdot H_k^{-1} g_k, \tag{17}$$

where

$$g_k = \nabla_\theta \mathbb{A}_c^{\theta_k}(\pi_\theta)|_{\theta=\theta_k} \quad \text{and} \quad H_k = \bar{H}_C(\theta_k) = \nabla_\theta^2 \bar{D}_C(\pi_\theta||\pi_{\theta_k})|_{\theta=\theta_k} \tag{18}$$

are finite sample estimates, and $H^{-1}g$ is approximated using conjugate gradients. The $\alpha^i \in [0,1]$ are the coefficients for backtracking line search, which ensures $\bar{D}_C(\pi_\theta||\pi_{\theta_k}) \leq \delta$.

We show in Appendix B.2.3 that the Hessian

$$\nabla_\theta^2 \bar{D}_C(\theta||\hat{\theta})|_{\theta=\hat{\theta}} = G_K(\theta_k) + \beta\phi''(b - V_c^{\hat{\theta}}(\theta))\nabla_\theta V_c^{\hat{\theta}}(\theta)^\top \nabla_\theta V_c^{\hat{\theta}}(\theta),$$

is equivalent to the Gramian $G_C(\theta_k)$ of the natural gradient update in Equation (19). We call the resulting policy gradient

$$\theta_{k+1} = \theta_k + \epsilon_k \bar{H}_C(\theta_k)^+ \nabla V_r(\theta_k), \tag{19}$$

the *Constrained NPG* (C-NPG). In particular, this shows that the C-TRPO update can be interpreted as a natural policy gradient step with an adaptive step size, see Appendix A. We emphasize that the idealized safe trust region update in Equation (11) and the C-TRPO update of Equation (13) agree up to second order in the policy parameters. This justifies the surrogate divergence in C-TRPO and motivates the discussion of the C-NPG flow in Section 4.2. We show in Theorem 5 that $\text{int}(\mathscr{D}_{\text{safe}})$ is invariant under the dynamics of the C-NPG. This implies that if the trust region radius $\delta$ is small, and the advantage estimation is accurate enough, the iterates under C-TRPO never leave the safe set.

## 4 Analysis

Here, we provide a theoretical analysis of the updates of C-TRPO and study the convergence properties of the time-continuous version of C-NPG. All proofs are deferred to the Appendix C.

## 4.1 Properties of the C-TRPO Update

The practical C-TRPO algorithm is implemented using the surrogate divergence introduced in Equation (13), which is identical to the theoretical divergence $D_C$ introduced in Equation (11) up to a mismatch between the policy advantage and the performance difference. The motivation for substituting the policy cost advantage for the performance difference is their equivalence up to first order and that we can estimate the advantage from samples of $d_\pi$. Similar to CPO, we can guarantee an almost improvement of the return (Achiam et al., 2017), despite the new divergence.

**Proposition 1** (C-TRPO reward update). *Set $\epsilon_r = \max_s |\mathbb{E}_{a \sim \pi_{k+1}} A_r^{\pi_k}(s, a)|$. The expected reward of a policy updated with C-TRPO is bounded from below by*

$$V_r(\pi_{k+1}) \geq V_r(\pi_k) - \frac{\sqrt{2\delta}\gamma\epsilon_r}{1-\gamma}. \tag{20}$$

Constraint violation, however, behaves slightly differently for the two algorithms. To see this, we establish a more concrete relation between C-TRPO and CPO. As $\beta \searrow 0$, the solution to Equation (13) approaches the constraint surface in the worst case, and we recover CPO, see Figure 1.

**Proposition 2.** *The approximate C-TRPO update approaches the CPO update in the limit as $\beta \searrow 0$.*

Intuitively, solving the C-TRPO problem with successively smaller values of $\beta$, would be similar to solving CPO with the interior point method using $\bar{D}_\phi(\cdot||\pi_k)$ as the barrier function.

Further, C-TRPO is more conservative than CPO for any $\beta > 0$ and as $\beta \to +\infty$ the updated is maximally constrained in the cost-increasing direction. This is formalized as follows.

**Proposition 3** (C-TRPO worst-case constraint violation). *Consider $\Psi \colon [0, \delta_b) \to [0, \infty)$ defined by $\Psi(x) = \phi(\delta_b - x) - \phi(\delta_b) - \phi'(\delta_b) \cdot x$ such that $D_\phi(\pi||\pi_k) = \Psi(\mathbb{A}_c^{\pi_k}(\pi))$. Further, set $\epsilon_c = \max_s |\mathbb{E}_{a \sim \pi_{k+1}} A_c^{\pi_k}(s, a)|$, and choose a strictly convex $\phi$. The worst-case constraint violation for C-TRPO is*

$$V_c(\pi_{k+1}) \leq V_c(\pi_k) + \Psi^{-1}(\delta/\beta) + \frac{\sqrt{2\delta}\gamma\epsilon_c}{1-\gamma}. \tag{21}$$

*Further, it holds that $\lim_{\beta \to +\infty} \Psi^{-1}(\delta/\beta) = 0$ and $\Psi^{-1}(\delta/\beta) < b - V_c(\pi_k)$ for all $\beta \in (0, \infty)$.*

This result is analogous to the worst-case constraint violation for CPO (Achiam et al., 2017, Proposition 2), except that it depends on the choice of $\beta$ and is tighter than the corresponding guarantee for CPO, because $\Psi^{-1}(\delta/\beta) < b - V_c(\pi_k)$ for all $\beta \in (0, \infty)$.

## 4.2 INVARIANCE AND CONVERGENCE OF CONSTRAINED NATURAL POLICY GRADIENTS

It is well known that TRPO is equivalent to a natural policy gradient method with an adaptive step size, see also Appendix A. We study the time-continuous limit of C-TRPO and guarantee safety during training and global convergence. In the context of constrained TRPO in Equation (11), we study the natural policy gradient flow

$$\partial_t \theta_t = G_{\mathrm{C}}(\theta_t)^+ \nabla V_r(\theta_t), \tag{22}$$

where $G_{\mathrm{C}}(\theta)^+$ denotes a pseudo-inverse of $G_{\mathrm{C}}(\theta)_{ij} = \partial_{\theta_i} d_\theta^\top \nabla^2 \Phi_{\mathrm{C}}(d_\theta) \partial_{\theta_j} d_\theta$ and $\theta \mapsto \pi_\theta$ is a differentiable policy parametrization. Moreover, we assume that $\theta \mapsto \pi_\theta$ is regular, that it is surjective and the Jacobian is of maximal rank everywhere. This assumption implies overparametrization but is satisfied for common models like tabular softmax, tabular escort, or expressive log-linear policy parameterizations (Agarwal et al., 2021a; Mei et al., 2020a; Müller & Montúfar, 2023).

We denote the set of safe parameters by $\Theta_{\mathrm{safe}} := \{\theta \in \mathbb{R}^p : \pi_\theta \in \Pi_{\mathrm{safe}}\}$, which is non-convex in general and say that $\Theta_{\mathrm{safe}}$ is *invariant* under Equation (22) if $\theta_0 \in \Theta_{\mathrm{safe}}$ implies $\theta_t \in \Theta_{\mathrm{safe}}$ for all $t$. Invariance is associated with safe control during optimization and is typically achieved via control barrier function methods (Ames et al., 2017; Cheng et al., 2019). We study the evolution of the state-action distributions $d_t = d^{\pi_{\theta_t}}$ as this allows us to employ the linear programming formulation of CMPDs and we obtain the following convergence guarantees.

**Theorem 4** (Safety during training). *Assume that $\phi \colon \mathbb{R}_{>0} \to \mathbb{R}$ satisfies $\phi'(x) \to +\infty$ for $x \searrow 0$ and consider a regular policy parameterization. Then the set $\Theta_{\mathrm{C}}$ is invariant under Equation (22).*

A visualization of policies obtained by C-NPG for different safe initializations and varying choices of $\beta$ is shown in Figure 2 for a toy MDP. We see that for even small choices of $\beta$ the trajectories don't cross the constraint surface and the updates become more conservative for larger choices of $\beta$.

**Theorem 5.** *Assume that $\phi'(x) \to +\infty$ for $x \searrow 0$, set $V_{r,\mathrm{C}}^\star := \max_{\pi \in \Pi_{\mathrm{safe}}} V_r(\pi)$ and denote the set of optimal constrained policies by $\Pi_{\mathrm{safe}}^\star = \{\pi \in \Pi_{\mathrm{safe}} : V_r(\pi) = V_{r,\mathrm{C}}^\star\}$, consider a regular policy parametrization and let $(\theta_t)_{t \geq 0}$ solve Equation (22). It holds that $V_r(\pi_{\theta_t}) \to V_{r,\mathrm{C}}^\star$ and*

$$\lim_{t \to +\infty} \pi_t = \pi_{\mathrm{safe}}^\star = \arg\min\{D_{\mathrm{C}}(\pi^\star, \pi_0) : \pi^\star \in \Pi_{\mathrm{safe}}^\star\}. \tag{23}$$

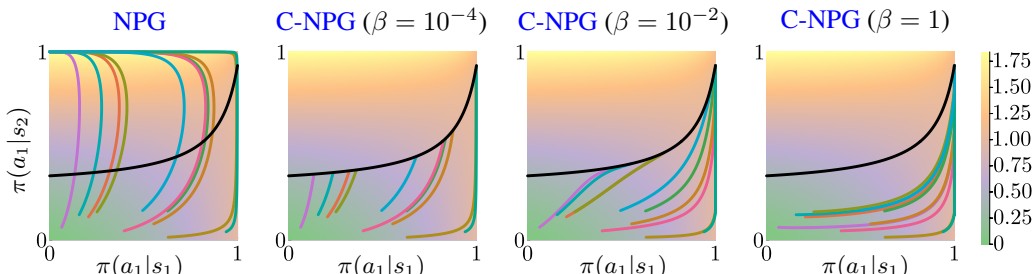

Figure 2: Shown is the policy set $\Pi \cong [0,1]^2$ for an MDP with two states and two actions with a heatmap of the reward $V_r$; the constraint surface is shown in black with the safe policies below; optimization trajectories are shown for 10 safe initialization and for $\beta = 0, 10^{-4}, 10^{-2}, 1$.

In case of multiple optimal policies, Equation (23) identifies the optimal policy of the CMDP that the natural policy gradient method converges to as the projection of the initial policy $\pi_0$ to the set of optimal safe policies $\Pi^\star_{\text{safe}}$ with respect to the constrained divergence $D_C$. In particular, this implies that the limiting policy $\pi^\star_{\text{safe}}$ satisfies as few constraints with equality as required to be optimal. To see this, note that $\Pi^\star_{\text{safe}}$ forms a face of $\mathscr{D}_{\text{safe}}$ and that Bregman projections lie at the interior of faces (Müller et al., 2024, Lemma A.2) and hence satisfy as few linear constraints as required.

## 5 COMPUTATIONAL EXPERIMENTS

**Setup and main results** We benchmark C-TRPO against 9 common safe policy optimization algorithms (CPO Achiam et al. (2017), PCPO Yang et al. (2020), CPPO-PID Stooke et al. (2020), PPO-Lag and TRPO-Lag Achiam et al. (2017); Ray et al. (2019), FOCOPS Zhang et al. (2020), CUP Yang et al. (2022), IPO Liu et al. (2020) and P3O Zhang et al. (2022)) on 8 tasks (4 Navigation and 4 Locomotion) from the Safety Gymnasium (Ji et al., 2023) benchmark. The locomotion tasks reward distance traveled, while penalizing high velocities, and the navigation tasks reward goal reaching and penalize certain unsafe states. For the C-TRPO implementation we fix the convex generator $\phi(x) = x \log(x)$, motivated by its superior performance in our experiments, see Appendix B.2.1, and $b_{\text{H}} = 0.8b$ and $\beta = 1$ across all experiments.[1] We train each algorithm for 10 million environment steps and evaluate on 10 runs after training, see Table 1 in Appendix D. Furthermore, each algorithm is trained with 5 seeds, and the cost regret is monitored throughout training for every run. To get a better sense of the safety of the algorithms during training, we take an online learning perspective and include as a metric the cumulative cost violation (Efroni et al., 2020; Müller et al., 2024)

$$\text{CUMCOST}_+(K, c) := \sum_{k=0}^{K-1} [V_c^{\pi_k} - b]_+ , \tag{24}$$

where $[x]_+ = \max\{0, x\}$, and $K$ is the number of the training iterations.

We observe that C-TRPO is competitive with the leading algorithms of the benchmark in terms of expected return (CPO, TRPO-Lagrangian), see Figure 3. Furthermore, it achieves notably lower cost regret throughout training than the high-return algorithms, even comparable to the more conservative PCPO algorithm. In Figure 3, we visualize the interquartile mean (IQM) of normalized scores across training for expected returns of reward and cost and for the cost regret, including their stratified bootstrap confidence intervals (Agarwal et al., 2021b).

**Discussion** For completeness, we also report environment-wise sample efficiency curves and evaluation performances in Appendix D.3. Our experiments reveal that the algorithm's performance is closely tied to the accuracy of divergence estimation, which hinges on the precise estimation of the cost advantage and value functions. The safety parameter $\beta$ modulates the stringency with which

---

[1]Code available at: (will be released after double-blind review)

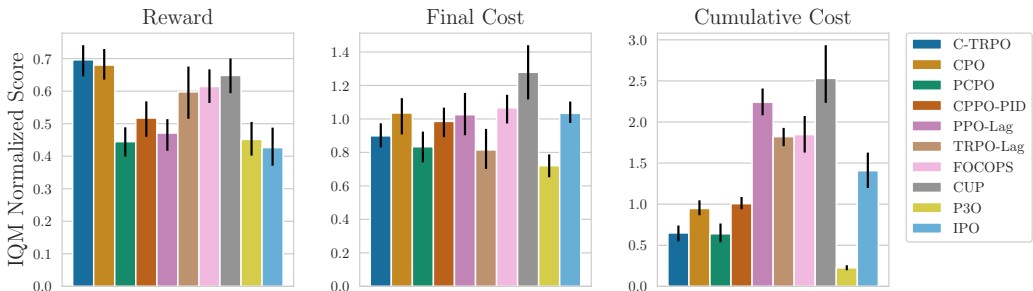

Figure 3: Comparison of safe policy optimization algorithms based on the Inter Quartile Mean across 5 seeds and 8 tasks. From left to right: episode return of the reward (PPO normalized), episode return of the cost (threshold normalized), and cumulative cost violation (CPO normalized).

C-TRPO satisfies the constraint, and can do so without limiting the expected return on most environments at least for $\beta \leq 1$, see Figure 7. For higher values, the expected return starts to degrade, partly due to $\bar{D}_\phi$ being relatively noisy compared to $\bar{D}_{\mathrm{KL}}$ and thus we recommend the choice $\bar{\beta} = 1$.

Further, we observe that constraint satisfaction is stable across different choices of cost threshold $b$, see Figure 8, and that in most environments, constraint violations seem to reduce as the algorithm converges, meaning that the regret flattens over time. This behavior suggests that the divergence estimation becomes increasingly accurate over time, potentially allowing C-TRPO to achieve sublinear regret. However, we leave regret analysis of the finite sample regime for future research.

We attribute the improved constraint satisfaction compared to CPO to a slowdown and reduction in the frequency of oscillations around the cost threshold, which mitigates overshoot behaviors that could otherwise violate constraints. The modified gradient preconditioner appears to deflect the parameter trajectory away from the constraint, see Figure 2. This effect may also be partially attributed to the hysteresis-based recovery mechanism, which helps smooth updates by leading the iterate away from the boundary of the safe set. Employing a hysteresis fraction $0 < b_{\mathrm{H}} < b$ might also be beneficial because C-TRPO's divergence estimates tend to be more reliable for strictly safe policies. The effect of the choice of $b_{\mathrm{H}}$ is shown in Figure 10 in the appendix. Finally, we present ablations in Appendix D.2, which support our claims that both components—the modified trust region and hysteresis—are effective in reducing safety violations.

## 6 CONCLUSION AND OUTLOOK

In this paper, we introduced C-TRPO and C-NPG, two novel methods for solving Constrained Markov Decision Processes (CMDPs). C-TRPO can be viewed as an extension or relaxation of Constrained Policy Optimization (CPO), from which a natural policy gradient method, C-NPG, is derived. C-TRPO represents a significant step toward safe, model-free reinforcement learning by integrating constraint handling directly into the geometry of the policy space. Meanwhile, C-NPG provides a provably safe natural policy gradient method for CMDPs, offering a foundational approach to direct policy optimization in constrained settings—similar to how NPG is a cornerstone in the theory of policy gradients for unconstrained MDPs. However, there are several limitations to address. First, the divergence estimation remains challenging, and we did not investigate the properties of the finite sample estimates of the divergence. In addition, the CMDP framework may be somewhat limited in modeling safe exploration and control. Because CMDPs constrain the *average* cost return, it can be difficult to model trajectory-wise or state-wise safety constraints. Several promising directions for future research remain open. One avenue is to combine these methods with model-based policy optimization to improve cost return estimates, or with policy mirror descent to improve computational efficiency, see e.g. Tomar et al. (2022). Additionally, integrating the proposed divergence with other safe policy optimization algorithms that utilize trust regions, e.g. PCPO, could lead to stronger performance guarantees.

Overall, the proposed algorithms, C-TRPO and C-NPG, present a step forward in general-purpose CMDP algorithms and move us closer to deploying RL in high-stakes, real-world applications.

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

## A   EXTENDED BACKGROUND

We consider the infinite-horizon discounted Markov decision process (MDP), given by the tuple $(\mathcal{S}, \mathcal{A}, P, r, \mu, \gamma)$. Here, $\mathcal{S}$ and $\mathcal{A}$ are the finite state-space and action-space respectively. Here, we make the restriction to finite MDPs as this simplifies the presentation. For a discussion of continuous state and action spaces, we refer to Appendix B.3. Further, $P\colon \mathcal{S} \times \mathcal{A} \to \Delta_{\mathcal{S}}$ is the transition kernel, $r\colon \mathcal{S} \times \mathcal{A} \to \mathbb{R}$ is the reward function, $\mu \in \Delta_{\mathcal{S}}$ is the initial state distribution at time $t = 0$, and $\gamma \in [0, 1)$ is the discount factor. The space $\Delta_{\mathcal{S}}$ is the set of categorical distributions over $\mathcal{S}$.

The Reinforcement Learning (RL) protocol is usually described as follows: At time $t = 0$, an initial state $s_0$ is drawn from $\mu$. At each integer time-step $t$, the agent chooses an action according to it's (stochastic) behavior policy $a_t \sim \pi(\cdot|s_t)$. A reward $r_t = r(s_t, a_t)$ is given to the agent, and a new state $s_{t+1} \sim P(\cdot|s_t, a_t)$ is sampled from the environment. Given a policy $\pi$, the value function $V_r^\pi\colon \mathcal{S} \to \mathbb{R}$, action-value function $Q_r^\pi\colon \mathcal{S} \times \mathcal{A} \to \mathbb{R}$, and advantage function $A_r^\pi\colon \mathcal{S} \times \mathcal{A} \to \mathbb{R}$ associated with the reward $r$ are defined as

$$V_r^\pi(s) \coloneqq (1 - \gamma)\, \mathbb{E}_\pi\left[\sum_{t=0}^\infty \gamma^t r(s_t, a_t)\Big| s_0 = s\right],$$

$$Q_r^\pi(s, a) \coloneqq (1 - \gamma)\, \mathbb{E}_\pi\left[\sum_{t=0}^\infty \gamma^t r(s_t, a_t)\Big| s_0 = s, a_0 = a\right] \text{ and } A_r^\pi(s, a) \coloneqq Q_r^\pi(s, a) - V_r^\pi(s).$$

where and the expectations are taken over trajectories of the Markov process resulting from starting at $s$ and following policy $\pi$. The goal is to

$$\text{maximize}_{\pi \in \Pi}\ V_r^\pi(\mu) \tag{25}$$

where $V_r^\pi(\mu)$ is the expected value under the initial state distribution $V_r^\pi(\mu) \coloneqq \mathbb{E}_{s \sim \mu}[V_r^\pi(s)]$. We will also write $V_r^\pi = V_r^\pi(\mu)$, and omit the explicit dependence on $\mu$ for convenience, and we write $V_r(\pi)$ when we want to emphasize its dependence on $\pi$.

**The Dual Linear Program for MDPs**   Any stationary policy $\pi$ induces a discounted state-action (occupancy) measure $d_\pi \in \Delta_{\mathcal{S} \times \mathcal{A}}$, indicating the relative frequencies of visiting a state-action pair, discounted by how far the visitation lies in the future. It is a probability measure defined as

$$d_\pi(s, a) \coloneqq (1 - \gamma) \sum_{t=0}^\infty \gamma^t \mathbb{P}_\pi(s_t = s)\pi(a|s), \tag{26}$$

where $\mathbb{P}_\pi(s_t = s)$ is the probability of observing the environment in state $s$ at time $t$ given the agent follows policy $\pi$. For finite MDPs, it is well-known that maximizing the expected discounted return can be expressed as the linear program

$$\max_d r^\top d \quad \text{subject to } d \in \mathscr{D}, \tag{27}$$

where $\mathscr{D}$ is the set of feasible state-action measures Feinberg & Shwartz (2012). This set is also known as the *state-action polytope*, defined by

$$\mathscr{D} = \left\{d \in \mathbb{R}_{\geq 0}^{\mathcal{S} \times \mathcal{A}} : \ell_s(d) = 0 \text{ for all } s \in \mathcal{S}\right\},$$

where the linear constraints $\ell_s(d)$ are given by the *Bellman flow equations*

$$\ell_s(d) = d(s) - \gamma \sum_{s', a'} d(s', a')P(s|s', a') - (1 - \gamma)\mu(s),$$

where $d(s) = \sum_a d(s, a)$ denotes the state-marginal of $d$. For any state-action measure $d$ we obtain the associated policy via conditioning, meaning

$$\pi(a|s) \coloneqq \frac{d(s, a)}{\sum_{a'} d(s, a')} \tag{28}$$

in case this is well-defined. This provides a one-to-one correspondence between policies and the state-action distributions under the following assumption.

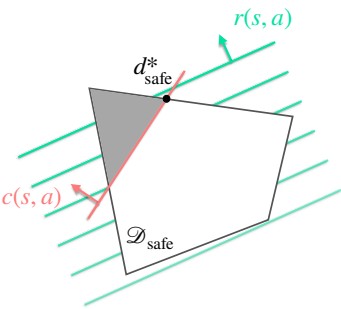

Figure 4: The dual linear program for a CMDP of two states and two actions.

**Assumption 6** (Exploration). For any policy $\pi \in \Delta_{\mathcal{A}}^{\mathcal{S}}$ we have $d_\pi(s) > 0$ for all $s \in \mathcal{S}$.

This assumption is standard in linear programming approaches and policy gradient methods where it is necessary for global convergence Kallenberg (1994); Mei et al. (2020b). Note that $d \in \partial \mathcal{D}$ if and only if $d(s, a) = 0$ for some $s, a$ and hence the boundary of $\mathcal{D}$ is given by

$$\partial \mathscr{D} = \Big\{ d_\pi : \pi(a|s) = 0 \text{ for some } s \in \mathcal{S}, a \in \mathcal{A} \Big\}.$$

**Constrained Markov Decision Processes**   Where MDPs aim to maximize the return, constrained MDPs (CMDPs) aim to maximize the return subject to a number of costs not exceeding certain thresholds. For a general treatment of CMDPs, we refer the reader to Altman (1999). An important application of CMDPs is in safety-critical reinforcement learning where the costs incorporate safety constraints. An infinite-horizon discounted CMDP is defined by the tuple $(\mathcal{S}, \mathcal{A}, P, r, \mu, \gamma, \mathcal{C})$, consisting of the standard elements of an MDP and an additional constraint set $\mathcal{C} = \{(c_i, b_i)\}_{i=1}^m$, where $c_i \colon \mathcal{S} \times \mathcal{A} \to \mathbb{R}$ are the cost functions and $b_i \in \mathbb{R}$ are the cost thresholds.

In addition to the value functions and the advantage functions of the reward that are defined for the MDP, we define the same quantities $V_{c_i}$, $Q_{c_i}$, and $A_{c_i}$ w.r.t the $i$th cost $c_i$, simply by replacing $r$ with $c_i$. The objective is to maximize the discounted return, as before, but we restrict the space of policies to the safe policy set

$$\Pi_{\text{safe}} = \bigcap_{i=1}^m \Big\{ \pi : V_{c_i}(\pi) \le b_i \Big\}, \tag{29}$$

where

$$V_{c_i}^\pi(\mu) \coloneqq \mathbb{E}_{s \sim \mu}[V_{c_i}^\pi(s)]. \tag{30}$$

is the expected discounted cumulative cost associated with the cost function $c_i$. Like the MDP, the discounted cost CMPD can be expressed as the linear program

$$\max_d r^\top d \quad \text{sbj. to } d \in \mathscr{D}_{\text{safe}}, \tag{31}$$

where

$$\mathscr{D}_{\text{safe}} = \bigcap_{i=1}^m \Big\{ d \in \mathbb{R}^{\mathcal{S} \times \mathcal{A}} : c_i^\top d \le b_i \Big\} \cap \mathscr{D} \tag{32}$$

is the safe occupancy set, see Figure 4.

**Bregman divergences**   Here, we give a short introduction to the concept of Bregman divergences, which is required for the formulation of trust region methods. For this, we consider a convex subset of Euclidean space $C \subseteq \mathbb{R}^d$ with a non-empty interior $\text{int}(C)$ and a strictly convex function $\phi \colon C \to \mathbb{R}$ which we assume to be differentiable on the interior $\text{int}(C)$. Then, the *Bregman divergence* induced by $\phi$ is given by

$$D_\phi(x||y) \coloneqq \phi(x) - \phi(y) - \nabla\phi(y)^\top (x - y), \tag{33}$$

which is well defined for $x \in C, y \in \text{int}(C)$. Intuitively, the Bregman divergence measures the difference between $\phi$ and its linearization at $y$. The strict convexity of $\phi$ ensures that $D_\phi(x||y) \geq 0$ and $D_\phi(x||y) = 0$ if and only if $x = y$. Therefore, Bregman divergences are commonly interpreted as a generalized measure for the distance between points, however, it is important to notice that it is not generally symmetric. An important example is the Euclidean distance $D_\phi(x||y) = \|x - y\|_2^2$ which arises from the choice $\phi(x) := \|x\|_2^2$. Another important Bregman divergence is the Kullback-Leibler (KL) divergence

$$D_{\text{KL}}(p||q) := \sum_{i=1}^{d} p_i \log \frac{p_i}{q_i} - \sum_{i=1}^{d} p_i + \sum_{i=1}^{d} q_i, \tag{34}$$

where we use the common convention $0 \log \frac{0}{0} := 0$. Then, the KL divergence is defined for $p \in \mathbb{R}_{\geq 0}^d$ and $q \in \mathbb{R}_{\geq 0}^d$ which is absolutely continuous with respect to $p$, meaning that $p_i = 0$ implies $q_i = 0$. Note that if both $p$ and $q$ are probability vectors, meaning that $\sum_i p_i = \sum_i q_i = 1$, we obtain

$$D_{\text{KL}}(p||q) := \sum_{i=1}^{d} p_i \log \frac{p_i}{q_i}. \tag{35}$$

**Information Geometry of Policy Optimization**   Among the most successful policy optimization schemes are natural policy gradient (NPG) methods or variants thereof like trust-region and proximal policy optimization (TRPO and PPO, respectively). These methods assume a convex geometry and corresponding Bregman divergences in the state-action polytope, where we refer to Neu et al. (2017); Müller & Montúfar (2023) for a more detailed discussion.

In general, a trust region update is defined as

$$\pi_{k+1} \in \underset{\pi \in \Pi}{\arg\max} \, \mathbb{A}_r^{\pi_k}(\pi) \quad \text{sbj. to } D_\Phi(d_{\pi_k}||d_\pi) \leq \delta, \tag{36}$$

where $D_\Phi \colon \mathscr{D} \times \mathscr{D} \to \mathbb{R}$ is a Bregman divergence induced by a suitably convex function $\Phi \colon \text{int}(\mathscr{D}) \to \mathbb{R}$. The functional

$$\mathbb{A}_r^{\pi_k}(\pi) = \mathbb{E}_{s \sim d_{\pi_k}, a \sim \pi_\theta(\cdot|s)} \big[ A_r^{\pi_k}(s, a) \big], \tag{37}$$

as introduced in (Kakade & Langford, 2002), is called the *policy advantage*. As a loss function, it is also known as the surrogate advantage (Schulman et al., 2017a), since we can interpret $\mathbb{A}$ as a surrogate optimization objective of the return. In particular, it holds for a parameterized policy $\pi_\theta$, that $\nabla_\theta \mathbb{A}_r^{\pi_{\theta_k}}(\pi_\theta)|_{\theta=\theta_k} = \nabla_\theta V_r(\theta_k)$, see Kakade & Langford (2002); Schulman et al. (2017a). TRPO and the original NPG assume the same geometry (Kakade, 2001; Schulman et al., 2017a), since they employ an identical Bregman divergence

$$D_{\text{K}}(d_{\pi_1}||d_{\pi_2}) := \sum_{s,a} d_{\pi_1}(s, a) \log \frac{\pi_1(a|s)}{\pi_2(a|s)} = \sum_s d_{\pi_1}(s) D_{\text{KL}}(\pi_1(\cdot|s)||\pi_2(\cdot|s)).$$

We refer to $D_{\text{K}}$ as the Kakade divergence and informally write $D_{\text{K}}(\pi_1, \pi_2) := D_{\text{K}}(d_{\pi_1}, d_{\pi_2})$. This divergence can be shown to be the Bregman divergence induced by the negative conditional entropy

$$\Phi_{\text{K}}(d_\pi) := \sum_{s,a} d_\pi(s, a) \log \pi(a|s), \tag{38}$$

see Neu et al. (2017). It is well known that with a parameterized policy $\pi_\theta$, a linear approximation of $\mathbb{A}$ and a quadratic approximation of the Bregman divergence $D_{\text{K}}$ at $\theta$, one obtains the *natural policy gradient* step given by

$$\theta_{k+1} = \theta_k + \epsilon_k G_{\text{K}}(\theta_k)^+ \nabla R(\theta_k), \tag{39}$$

where $G_{\text{K}}(\theta)^+$ denotes a pseudo-inverse of the Gramian matrix with entries equal to the state-averaged Fisher-information matrix of the policy

$$G_{\text{K}}(\theta)_{ij} := \mathbb{E}_{s \sim d_{\pi_\theta}} \left[ \sum_a \frac{\partial_{\theta_i} \pi_\theta(a|s) \partial_{\theta_j} \pi_\theta(a|s)}{\pi_\theta(a|s)} \right] \tag{40}$$

$$= \mathbb{E}_{d_{\pi_\theta}} [\partial_{\theta_i} \log \pi_\theta(a|s) \partial_{\theta_j} \log \pi_\theta(a|s)], \tag{41}$$

where we refer to Schulman et al. (2017a) for a more detailed discussion.

Consider a convex potential $\Phi\colon \mathscr{D} \to \mathbb{R}$ or $\Phi\colon \mathscr{D}_{\mathrm{safe}} \to \mathbb{R}$ and the TRPO update

$$\theta_{k+1} \in \arg\max \mathbb{A}_r^{\pi_{\theta_k}}(\pi_\theta) \quad \text{sbj. to } D_\Phi(d_{\theta_k}||d_\theta) \leq \epsilon. \tag{42}$$

In practice, one uses a linear approximation of $\mathbb{A}_r^{\pi_{\theta_k}}(\pi_\theta)$ and a quadratic approximation of $D_\Phi$ to compute the TRPO update. This gives the following approximation of TRPO

$$\theta_{k+1} \in \arg\max_\theta \nabla_\theta \mathbb{A}_r^{\theta_k}(\theta)|_{\theta=\theta_k} \cdot (\theta - \theta_k) \quad \text{sbj. to } \|\theta - \theta_k\|_{G(\theta_k)}^2 \leq \epsilon, \tag{43}$$

where

$$G(\theta)_{ij} = \partial_{\theta_i} d_\theta^\top \nabla^2 \Phi(d_\theta) \partial_{\theta_j} d_\theta. \tag{44}$$

Note that by the policy gradient theorem, it holds that

$$\nabla_\theta \mathbb{A}_r^{\theta_k}(\theta)|_{\theta=\theta_k} = \nabla V_r(\theta_k). \tag{45}$$

Thus, the approximate TRPO update is equivalent to

$$\theta_{k+1} = \theta_k + \epsilon_k G(\theta_k)^+ \nabla V_r(\theta), \tag{46}$$

where

$$\epsilon_k = \frac{\sqrt{\epsilon}}{\|G(\theta_k)^+ \nabla V_r(\theta_k)\|_{G(\theta_k)}}. \tag{47}$$

Hence, the approximation TRPO update corresponds to a natural policy gradient update with an adaptively chosen step size.

# B DETAILS ON THE SAFE GEOMETRY FOR CMDPs

## B.1 SAFE TRUST REGIONS

The safe mirror function for a single constraint is given by

$$\Phi_{\mathrm{C}}(d) := \Phi_{\mathrm{K}}(d) + \sum_{i=1}^m \beta\,\phi(b - c^\top d), \tag{48}$$

and the resulting Bregman divergence

$$D_{\mathrm{C}}(d_1||d_2) = \Phi_{\mathrm{C}}(d_1) - \Phi_{\mathrm{C}}(d_2) - \langle \nabla\Phi_{\mathrm{C}}(d_2), d_1 - d_2 \rangle. \tag{49}$$

is a linear operator in $\Phi$, hence

$$D_{\Phi(d)+\beta\phi(b-c^\top d)}(d_1||d_2) = D_{\Phi_{\mathrm{K}}}(d_1||d_2) + \beta D_\phi(d_1||d_2), \tag{50}$$

where

$$D_\phi(d_1||d_2) = \phi(b - c^\top d_1) - \phi(b - c^\top d_2) - \langle \nabla\phi(b - c^\top d_2), d_1 - d_2 \rangle \tag{51}$$

$$= \phi(b - c^\top d_1) - \phi(b - c^\top d_2) - \phi'(b - c^\top d_2)(c^\top d_1 - c^\top d_2). \tag{52}$$

$$= \phi(b - V_c(\pi)) - \phi(b - V_c(\pi_k)) + \phi'(b - V_c(\pi_k))(V_c(\pi) - V_c(\pi_k)). \tag{53}$$

The last expression can be interpreted as the one-dimensional Bregman divergence $D_\phi(b - V_c(\pi)||b - V_c(\pi_k))$, which is a (strictly) convex function in $V_c(\pi)$ for fixed $\pi_k$ if $\phi$ is (strictly) convex.

## B.2 DETAILS ON C-TRPO

### B.2.1 SURROGATE DIVERGENCE

In practice, the exact constrained KL-Divergence $D_{\mathrm{C}}$ cannot be evaluated, because it depends on the cost-return of the optimized policy $V_c(\pi)$. Therefore, we use the surrogate divergence

$$\bar{D}_\phi(\pi_\theta||\pi_{\theta_k}) = \phi(b - V_c^{\pi_k} - \mathbb{A}_c^{\pi_k}(\pi)) - \phi(b - V_c^{\pi_k}) + \phi'(b - V_c^{\pi_k})\mathbb{A}_c^{\pi_k}(\pi) \tag{54}$$

is obtained by the substitution $V_c(\pi) - V_c^{\pi_k} \to \mathbb{A}_c^{\pi_k}(\pi)$ in $D_\phi$.

When we center this divergence around policy $\pi_k$ and keep this policy fixed, it becomes a function of the policy cost advantage.

$$
\begin{aligned}
\bar{D}_\phi(\pi_\theta || \pi_{\theta_k}) &= \phi(b - V_c^{\pi_k} - \mathbb{A}_c^{\pi_k}(\pi)) - \phi(b - V_c^{\pi_k}) + \phi'(b - V_c^{\pi_k})\mathbb{A}_c^{\pi_k}(\pi) \\
&= \phi(\delta_b - \mathbb{A}_c^{\pi_k}(\pi)) - \phi(\delta_b) + \phi'(\delta_b)\mathbb{A}_c^{\pi_k}(\pi) \\
&= \Psi(\mathbb{A}_c^{\pi_k}).
\end{aligned}
$$

Note that $\bar{D}_\phi(\pi_\theta || \pi_{\theta_k}) = \Psi(\mathbb{A}_c^{\pi_k}(\pi))$, where $\Psi(x) = \phi(\delta_b - x) - \phi(\delta_b) - \phi'(\delta_b) \cdot x$ is a (strictly) convex function if $\phi$ is (strictly) convex, since it is equivalent to the one-dimensional Bregman divergence $D_\phi(\delta_b - x || \delta_b)$ on the domain of $\phi(b - x)$, see Figure 5.

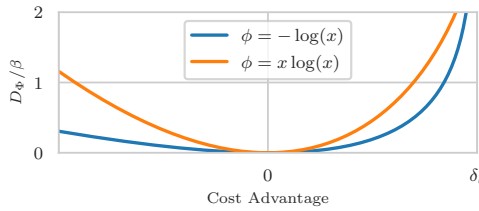

Figure 5: The surrogate Constrained KL-Divergence as a function of the policy cost advantage.

**Example 7.** The function $\phi(x) = x \log(x)$ induces the divergence

$$
\bar{D}_\phi(\pi_\theta || \pi_{\theta_k}) = \mathbb{A}_c^{\pi_k}(\pi_\theta) - (\delta_b - \mathbb{A}_c^{\pi_k}(\pi_\theta)) \log \left( \frac{\delta_b}{\delta_b - \mathbb{A}_c^{\pi_k}(\pi_\theta))} \right). \tag{55}
$$

### B.2.2 ESTIMATION

In the practical implementation, the expected KL-divergence between the policy of the previous iteration, $\pi_k$, and the proposal policy $\pi$ is estimated from state samples $s_i$ by running $\pi_k$ in the environment

$$
\sum_s d_{\pi_k}(s) D_{\mathrm{KL}}(\pi(\cdot|s) || \pi_k(\cdot|s)) \approx 1/N \sum_{i=0}^{N-1} D_{\mathrm{KL}}(\pi(\cdot|s_i) || \pi_k(\cdot|s_i)) \tag{56}
$$

where $D_{\mathrm{KL}}$ can be computed in closed form for Gaussian policies, where $N$ is the batch size.

For the constraint term, we estimate $\delta_b$ from trajectory samples, as well as the policy cost advantage

$$
\mathbb{A}_c^{\pi_k}(\pi) \approx \hat{\mathbb{A}} = \frac{1}{N} \sum_{i=0}^{N-1} \frac{\pi(a_i|s_i)}{\pi_k(a_i|s_i)} \hat{A}_i^{\pi_k} \tag{57}
$$

where $\hat{A}_i^{\pi_k}$ is the GAE-$\lambda$ estimate of the advantage function (Schulman et al., 2018). For any suitable $\phi$, the resulting divergence estimate is

$$
\hat{D}_\phi = \phi(\delta_b - \hat{\mathbb{A}}) - \phi(\delta_b) - \phi'(\delta_b)\hat{\mathbb{A}} \tag{58}
$$

and for the specific choice $\phi(x) = x \log(x)$

$$
\hat{D}_\phi = \hat{\mathbb{A}} - (\delta_b - \hat{\mathbb{A}}) \log \left( \frac{\delta_b}{\delta_b - \hat{\mathbb{A}}} \right). \tag{59}
$$

### B.2.3 DETAILS ON C-NPG

In showin that TRPO with quadratic approximation agrees with a natural gradient step, see Appendix A, we have used that $\nabla_\theta \mathbb{A}_r^{\theta_k}(\theta)|_{\theta=\theta_k} = \nabla V_r(\theta_k)$, which holds although $\mathbb{A}_r$ is only a proxy of $V_r$. We now provide a similar property for the quadratic approximation of the surrogate divergences $\bar{D}_C$.

**Proposition 8.** *For any parameter $\theta$ with $\pi_\theta \in \Pi_{\text{safe}}$ it holds that*

$$\nabla_\theta^2 \bar{D}_\phi(\theta||\hat{\theta})|_{\theta=\hat{\theta}} = \nabla_\theta^2 D_\phi(\theta||\hat{\theta})|_{\theta=\hat{\theta}} \tag{60}$$

*and hence*

$$\nabla_\theta^2 \bar{D}_{\text{KL}}(\theta||\hat{\theta})|_{\theta=\hat{\theta}} + \beta\nabla_\theta^2 \bar{D}_\phi(\theta||\hat{\theta})|_{\theta=\hat{\theta}} = G_C(\hat{\theta}) \tag{61}$$

*where $G_C(\theta)$ denotes the Gramian matrix of C-NPG with entries*

$$G_C(\theta)_{ij} = \partial_{\theta_i} d_\theta^\top \nabla^2 \Phi_C(\theta) \partial_{\theta_j} d_\theta. \tag{62}$$

*Proof.* Let $\bar{H}_{\text{KL}}(\theta) = \nabla_\theta^2 \bar{D}_{\text{KL}}(\theta||\hat{\theta})|_{\theta=\hat{\theta}}$ and $\bar{H}_\phi(\theta) = \nabla_\theta^2 \bar{D}_\phi(\theta||\hat{\theta})|_{\theta=\hat{\theta}}$. One can show that $\bar{H}_{\text{KL}} = G_K(\theta)$ (Schulman et al., 2017a). Further, we have

$$\bar{H}_\phi(\theta) = \nabla_\theta \mathbb{A}_c^{\pi_k}(\theta)^\top \Psi''(\mathbb{A}_c^{\pi_k}(\theta))\nabla_\theta \mathbb{A}_c^{\pi_k}(\theta) + \Psi'(\mathbb{A}_c^{\pi_k}(\theta))^\top \nabla_\theta^2 \mathbb{A}_c^{\pi_k}(\theta)$$

$$\overset{a)}{=} \nabla_\theta \mathbb{A}_c^{\pi_k}(\theta)^\top \Psi''(\mathbb{A}_c^{\pi_k}(\theta))\nabla_\theta \mathbb{A}_c^{\pi_k}(\theta)$$

$$\overset{b)}{=} \nabla_\theta \mathbb{A}_c^{\pi_k}(\theta)^\top \phi''(b - V_c^{\pi_k}(\theta))\nabla_\theta \mathbb{A}_c^{\pi_k}(\theta)$$

$$= \nabla_\theta V_c^{\pi_k}(\theta)^\top \phi''(b - V_c^{\pi_k}(\theta))\nabla_\theta V_c^{\pi_k}(\theta),$$

where a) follows from $\Psi'(\mathbb{A}_c^{\pi_k}(\theta)) = 0$ since $\Psi(0) = 0$, $\Psi \geq 0$ and $\mathbb{A}_c^{\hat{\theta}}(\theta)|_{\theta=\hat{\theta}} = 0$. Further, b) follows because $\Psi''(x)|_{x=0} = \phi''(\delta_b)$. Thus, $\bar{H}_\phi$ is equivalent to the Gramian

$$G_C(\theta)_{ij} := \partial_{\theta_i} d_\theta^\top \nabla^2 \Phi_C(\theta) \partial_{\theta_j} d_\theta \tag{63}$$

$$= G_K(\theta)_{ij} + \beta\phi''(b - c_k^\top d_\theta)\partial_{\theta_i} d_\theta^\top cc^\top \partial_{\theta_i} d_\theta \tag{64}$$

$$= \bar{H}_{\text{KL}} + \beta\nabla_\theta V_c(\theta)^\top \phi''(b - V_c(\theta))\nabla_\theta V_c(\theta), \tag{65}$$

$$= \bar{H}_{\text{KL}} + \beta\bar{H}_\phi. \tag{66}$$

Again, for multiple constraints, the statement follows analogously. □

In particular, this shows that the C-TRPO update can be interpreted as a natural policy gradient step with an adaptive step size and that the updates with $D_C$ and $\bar{D}_C$ are equivalent if we use a quadratic approximation for both, justifying $\bar{D}_C$ as a surrogate for $D_C$.

### B.3 BEYOND FINITE MDPs

For the sake of simplicity and as this is required for our theoretical analysis, we have introduced C-TRPO only for finite MDPs. However, C-TRPO can also be used for problems with continuous state and action spaces as we discuss here. In this case, the state-action and state distributions are defined as

$$d_\pi(S \times A) := (1 - \gamma)\sum_{t=0}^\infty \gamma^t \mathbb{P}_\pi(s_t \in S, a_t \in A) \quad \text{and}$$

$$d_\pi(S) := (1 - \gamma)\sum_{t=0}^\infty \gamma^t \mathbb{P}_\pi(s_t \in S)$$

for every measurable subsets $A \subseteq \mathcal{A}$ and $S \subseteq \mathcal{S}$. Further, the Kakade divergence is then given by

$$D_K(d^{\pi_1}||d^{\pi_2}) := \mathbb{E}_{s \sim d^{\pi_1}}\left[D_{\text{KL}}(\pi_1(\cdot|s)||\pi_2(\cdot|s))\right], \tag{67}$$

which is well defined if $\pi_1(\cdot|s)$ is absolutely continuous with respect to $\pi_2(\cdot|s)$ for $d^{\pi_1}$ almost all $s \in \mathcal{S}$. The Bregman divergence that C-TRPO is builds on is – just as in the finite case – given by

$$D_{\mathrm{C}}(d_1||d_2) = D_{\mathrm{K}}(d_1||d_2) + \sum_{i=1}^{m} \beta_i D_{\phi_i}(d_1||d_2), \tag{68}$$

where

$$D_{\phi_i}(d_1||d_2) = \phi(b_i - V_{c_i}(\pi_1)) - \phi(b_i - V_{c_i}(\pi_2)) + \phi'(b_i - V_{c_i}(\pi_2))(V_{c_i}(\pi_1) - V_{c_i}(\pi_2)). \tag{69}$$

Like in the finite case, the policy advantage is defined as

$$\mathbb{A}_r^{\pi_k}(\pi) = \mathbb{E}_{s,a \sim d_{\pi_k}} \left[ \frac{\pi(a|s)}{\pi_k(a|s)} A_r^{\pi_k}(s,a) \right], \tag{70}$$

where $A_r^\pi(s,a) = Q^\pi(s,a) - V^\pi(s)$ denotes the advantage function, which is defined analoguously to the finite case. Now, the plain trust region update is given b y

$$\theta_{k+1} \in \arg\max_\theta \mathbb{A}_r^{\pi_k}(\pi) \quad \text{sbj. to } D_{\mathrm{C}}(d_{\pi_k}||d_\pi) \le \delta. \tag{71}$$

Just like in the finite case, we use a surrogate divergence $\bar{D}_{\mathrm{C}}$ and obtain the formulation of C-TRPO

$$\pi_{k+1} = \arg\max_{\pi \in \Pi} \mathbb{A}_r^{\pi_k}(\pi) \quad \text{sbj. to } \bar{D}_{\mathrm{C}}(\pi||\pi_k) \le \delta. \tag{72}$$

Here, the differences to $D_{\mathrm{C}}$ are that we use use samples from the state distribution $d^{\pi_k}$ and use a surrogate for the cost advantage to estimate the divergence $D_{\pi_i}$ as described in Section 3.2. Further, we use a parametric policy model $\pi_\theta$ and a linear approximation of $\mathbb{A}^{\pi_k}$ as well as quadratic approximation of $\bar{D}_{\mathrm{C}}(\pi||\pi_k)$ for our practical implementation.

**Expression for Gaussian policies**    We test C-TRPO in various control tasks and hence, where we use Gaussian policies. More precisely, the state and action space consist of Euclidean spaces $\mathcal{S} = \mathbb{R}^{d_s}$ and $\mathcal{A} = R^{d_a}$. Then, we consider a policy network $\mu_\theta \colon \mathcal{S} \to \mathcal{A}$, which predicts the mean action and assume parameterized but state independent diagonal Gaussian noise, meaning that $\pi_\theta(\cdot|s) = \mathcal{N}(\mu_\theta(s), \Sigma_\theta)$, where $\Sigma_\theta$ is diagonal. Consequently, we can use a closed-form expression for the KL divergence as

$$D_{\mathrm{KL}}(\pi_{\theta_1}(\cdot|s)||\pi_{\theta_2}(\cdot|s)) = \frac{1}{2}\left(\mathrm{tr}\left(\Sigma_{\theta_2}^{-1}\Sigma_{\theta_1}\right) - d_a + \|\mu_{\theta_1}(s) - \mu_{\theta_2}(s)\|_{\Sigma_{\theta_2}^{-1}}^2 + \ln\left(\frac{\det \Sigma_{\theta_2}}{\det \Sigma_{\theta_1}}\right)\right),$$

see Zhang et al. (2024b).

# C    PROOFS OF SECTION 4

## C.1    PROOFS OF SECTION 4.1

Our theoretical analysis of C-TRPO is built on the following bounds on the performance difference of two policies.

**Theorem 9** (Performance Difference, Achiam et al. (2017)). *For any function $f(s,a)$, the following bounds hold*

$$V_f(\pi_1) - V_f(\pi_2) \lesseqgtr \mathbb{A}_f^{\pi_2}(\pi_1) \pm \frac{2\gamma\epsilon_f}{(1-\gamma)}\sqrt{\frac{1}{2}\mathbb{E}_{s \sim d_{\pi_2}} D_{\mathrm{KL}}(\pi_1(\cdot|s)||\pi_2(\cdot|s))} \tag{73}$$

*where $\epsilon_f = \max_s |\mathbb{E}_{a \sim \pi_1} A_f^{\pi_2}(s,a)|$.*

Theorem 9 can be interpreted as a bound on the error incurred by replacing the difference in returns $V_f(\pi_1) - V_f(\pi)$ of any state-action function by its policy advantage $\mathbb{A}_f^{\pi_2}(\pi_1)$.

**Proposition 1** (C-TRPO reward update). *Set $\epsilon_r = \max_s |\mathbb{E}_{a \sim \pi_{k+1}} A_r^{\pi_k}(s,a)|$. The expected reward of a policy updated with C-TRPO is bounded from below by*

$$V_r(\pi_{k+1}) \ge V_r(\pi_k) - \frac{\sqrt{2\delta}\gamma\epsilon_r}{1-\gamma}. \tag{20}$$

*Proof.* It follows from the lower bound in Theorem 9 that

$$V_r(\pi_{k+1}) - V_r(\pi_k) \geq \mathbb{A}_r^{\pi_k}(\pi_{k+1}) - \frac{\gamma\epsilon_r}{(1-\gamma)}\sqrt{2\bar{D}_C(\pi_{k+1}||\pi_k)} \tag{74}$$

where we choose $f = r$. The bound holds because $\bar{D}_\phi \geq 0$, and thus $\bar{D}_C \geq \mathbb{E}D_{KL}$. Further, $\delta \geq D_C$ and $\mathbb{A}_r^{\pi_k}(\pi_{k+1}) \geq 0$ by the update equation, which concludes the proof. See Appendix C.3 for a more detailed discussion. $\square$

**Proposition 2.** *The approximate C-TRPO update approaches the CPO update in the limit as $\beta \searrow 0$.*

*Proof.* Let us fix a strictly safe policy $\pi_0 \in \text{int}(\Pi_{\text{safe}})$. In both cases, we approximate the expected cost of a policy using $V_c(\pi) \approx V_c(\pi_0) + \mathbb{A}_c^{\pi_0}(\pi)$, which is off by the advantage mismatch term in Theorem 1. Hence, we maximize the surrogate of the expected value $\mathbb{A}_r^{\pi_0}(\pi)$ over the regions

$$P_{\text{CPO}} := \{\pi \in \Pi : \bar{D}_K(\pi, \pi_0) \leq \delta, V_c(\pi_0) + \mathbb{A}_c^{\pi_0}(\pi) \leq b\}$$

in the case of CPO, and

$$P_\beta := \{\pi \in \Pi : \bar{D}_C(\pi, \pi_0) \leq \delta\},$$

with C-TRPO for some $\beta > 0$. Note that

$$\bar{D}_C(\pi, \pi_0) = \bar{D}_K(\pi, \pi_0) + \beta\Psi(\mathbb{A}_c^{\pi_0}(\pi)), \tag{75}$$

and $\Psi : (-\infty, \delta_b) \to (0, +\infty)$ and $\Psi(t) \to +\infty$ for $t \nearrow \delta_b$, where $\delta_b = b - V_c(\pi_0)$. Denote the corresponding updates by $\hat{\pi}_{\text{CPO}}$ and the C-TRPO update by $\hat{\pi}_\beta$. Note that we have $P_\beta \subseteq P_{\beta'} \subseteq P_{\text{CPO}}$ for $\beta \geq \beta'$. Further, we have

$$\bigcup_{\beta>0} P_\beta = \{\pi \in P : D_K(\pi, \pi_0) < \delta, V_c(\pi_0) + \mathbb{A}_c^{\pi_0}(\pi) < b\}.$$

Hence, the trust regions $P_\beta$ grow for $\beta \searrow 0$ and fill the interior of the trust region $P_{\text{CPO}}$. $\square$

*Remark* 10. Intuitively, one could repeatedly solve the C-TRPO problem with successively smaller values of $\beta$, which would be similar to solving CPO with the interior point method using $\Psi$ as the barrier function.

**Proposition 3** (C-TRPO worst-case constraint violation). *Consider $\Psi : [0, \delta_b) \to [0, \infty)$ defined by $\Psi(x) = \phi(\delta_b - x) - \phi(\delta_b) - \phi'(\delta_b) \cdot x$ such that $D_\phi(\pi||\pi_k) = \Psi(\mathbb{A}_c^{\pi_k}(\pi))$. Further, set $\epsilon_c = \max_s |\mathbb{E}_{a\sim\pi_{k+1}} A_c^{\pi_k}(s, a)|$, and choose a strictly convex $\phi$. The worst-case constraint violation for C-TRPO is*

$$V_c(\pi_{k+1}) \leq V_c(\pi_k) + \Psi^{-1}(\delta/\beta) + \frac{\sqrt{2\delta}\gamma\epsilon_c}{1-\gamma}. \tag{21}$$

*Further, it holds that $\lim_{\beta\to+\infty}\Psi^{-1}(\delta/\beta) = 0$ and $\Psi^{-1}(\delta/\beta) < b - V_c(\pi_k)$ for all $\beta \in (0, \infty)$.*

*Proof.* Setting $f = c$ in the upper bound from Theorem 9, and replacing $\mathbb{E}D_{KL}$ with $\delta$ as in Proposition 1 results in

$$V_c(\pi_{k+1}) \leq V_c(\pi_k) + \mathbb{A}_c^{\pi_k}(\pi_{k+1}) + \frac{\sqrt{2\delta}\gamma\epsilon_c}{1-\gamma}. \tag{76}$$

Recall that $\bar{D}_C = \bar{D}_K + \beta\bar{D}_\phi$ and that $\bar{D}_\phi(\pi_{k+1}||\pi_k) = \Psi(\mathbb{A}_c^{\pi_k}(\pi_{k+1}))$, where $\Psi(x) = \phi(\delta_b - x) - \phi(\delta_b) - \phi'(\delta_b) \cdot x$. By the definition of the update it holds that

$$\Psi(\mathbb{A}_c^{\pi_k}(\pi_{k+1})) < \delta/\beta. \tag{77}$$

Since we are only interested in upper bounding the worst case, we can focus on $\mathbb{A}_c^{\pi_k}(\pi_{k+1}) > 0$, so we restrict $\Psi : [0, \delta_b) \to [0, \infty)$. Further, for strictly convex $\phi$, $\Psi$ is strictly convex and increasing with increasing inverse. It follows that

$$\mathbb{A}_c^{\pi_k}(\pi_{k+1}) < \Psi^{-1}(\delta/\beta), \tag{78}$$

with $\Psi^{-1} : [0, \infty) \to [0, \delta_b)$. Because $\Psi^{-1}$ is an increasing function of $\beta$ on $[0, \infty)$ with maximum at $\delta_b = b - V_c(\pi_k)$, it holds that $\Psi^{-1}(\beta/\delta) < b - V_c(\pi_k)$ for any $\beta > 0$, which concludes the proof. $\square$

## C.2 DETAILS ON THE RESULTS IN SECTION 4.2

Recall that we study the natural policy gradient flow

$$\partial_t \theta_t = G_{\mathrm{C}}(\theta_t)^+ \nabla V_r(\theta_t), \tag{79}$$

where $G_{\mathrm{C}}(\theta)^+$ denotes a pseudo-inverse of $G_{\mathrm{C}}(\theta)$ with entries

$$G_{\mathrm{C}}(\theta)_{ij} := \partial_{\theta_i} d_\theta^\top \nabla^2 \Phi_{\mathrm{C}}(d_\theta) \partial_{\theta_j} d_\theta = G_{\mathrm{K}}(\theta)_{ij} + \sum_k \beta_k \phi''(b_k - c_k^\top d_\theta) \partial_{\theta_i} d_\theta^\top c_k c_k^\top \partial_{\theta_i} d_\theta. \tag{80}$$

and $\theta \mapsto \pi_\theta$ is a differentiable policy parametrization.

Moreover, we assume that $\theta \mapsto \pi_\theta$ is regular, that it is surjective and the Jacobian is of maximal rank everywhere. This assumption implies overparametrization but is satisfied for common models like tabular softmax, tabular escort, or expressive log-linear policy parameterizations (Agarwal et al., 2021a; Mei et al., 2020a; Müller & Montúfar, 2023).

We denote the set of safe parameters by $\Theta_{\mathrm{safe}} := \{\theta \in \mathbb{R}^p : \pi_\theta \in \Pi_{\mathrm{safe}}\}$, which is non-convex in general and say that $\Theta_{\mathrm{safe}}$ is *invariant* under Equation (22) if $\theta_0 \in \Theta_{\mathrm{safe}}$ implies $\theta_t \in \Theta_{\mathrm{safe}}$ for all $t$. Invariance is associated with safe control during optimization and is typically achieved via control barrier function methods (Ames et al., 2017; Cheng et al., 2019). We study the evolution of the state-action distributions $d_t = d^{\pi_{\theta_t}}$ as this allows us to employ the linear programming formulation of CMPDs and we obtain the following convergence guarantees.

**Theorem 4** (Safety during training). *Assume that $\phi \colon \mathbb{R}_{>0} \to \mathbb{R}$ satisfies $\phi'(x) \to +\infty$ for $x \searrow 0$ and consider a regular policy parameterization. Then the set $\Theta_{\mathrm{C}}$ is invariant under Equation (22).*

*Proof.* Consider a solution $(\theta_t)_{t>0}$ of Equation (79). As the mapping $\pi \mapsto d^\pi$ is a diffeomorphism (Müller & Montúfar, 2023) the parameterization $\Theta_{\mathrm{safe}} \to \mathscr{D}_{\mathrm{safe}}, \theta \mapsto d^{\pi_\theta}$ is surjective and has a Jacobian of maximal rank everywhere. As $G_{\mathrm{C}}(\theta)_{ij} = \partial_{\theta_i} d_\theta \nabla \Phi_{\mathrm{C}} \partial_{\theta_i} d_\theta$ this implies that the state-action distributions $d_t = d^{\pi_{\theta_t}}$ solve the Hessian gradient flow with Legendre-type function $\Phi_{\mathrm{C}}$ and the linear objective $d \mapsto r^\top d$, see Amari (2016); van Oostrum et al. (2023); Müller & Montúfar (2023) for a more detailed discussion. It suffices to study the gradient flow in the space of state-action distributions $d_t$. It is easily checked that $\Phi_{\mathrm{C}}$ is a Legendre-type function for the convex domain $\mathscr{D}_{\mathrm{C}}$, meaning that it satisfies $\|\nabla \Phi(d_n)\| \to +\infty$ for $d_n \to d \in \partial \mathscr{D}_{\mathrm{safe}}$. Since the objective is linear, it follows from the general theory of Hessian gradient flows of convex programs that the flow is well posed, see Alvarez et al. (2004); Müller & Montúfar (2023). $\square$

**Theorem 5.** *Assume that $\phi'(x) \to +\infty$ for $x \searrow 0$, set $V_{r,\mathrm{C}}^\star := \max_{\pi \in \Pi_{\mathrm{safe}}} V_r(\pi)$ and denote the set of optimal constrained policies by $\Pi_{\mathrm{safe}}^\star = \{\pi \in \Pi_{\mathrm{safe}} : V_r(\pi) = V_{r,\mathrm{C}}^\star\}$, consider a regular policy parametrization and let $(\theta_t)_{t\geq 0}$ solve Equation (22). It holds that $V_r(\pi_{\theta_t}) \to V_{r,\mathrm{C}}^\star$ and*

$$\lim_{t \to +\infty} \pi_t = \pi_{\mathrm{safe}}^\star = \arg\min\{D_{\mathrm{C}}(\pi^\star, \pi_0) : \pi^\star \in \Pi_{\mathrm{safe}}^\star\}. \tag{23}$$

*Proof.* Just like in the proof of Theorem 5 we see that $d_t = d^{\pi_{\theta_t}}$ solves the Hessian gradient flow with respect to the Legendre type function $\Phi_{\mathrm{C}}$. Now the claims regarding convergence and the identification of the limit $\lim_{t \to +\infty} \pi_{\theta_t}$ follows from the general theory of Hessian gradient flows, see Alvarez et al. (2004); Müller et al. (2024). $\square$

## C.3 PERFORMANCE IMPROVEMENT BOUNDS AND CHOICE OF DIVERGENCE

In a series of works (Kakade & Langford, 2002; Pirotta et al., 2013; Schulman et al., 2017a; Achiam et al., 2017), the following bound on policy performance difference between two policies has been established.

$$V_f(\pi') - V_f(\pi) \lesseqqgtr \mathbb{A}_f^{\pi'}(\pi) \pm \frac{2\gamma\epsilon_f}{(1-\gamma)} \mathbb{E}_{s \sim d_\pi} D_{\mathrm{TV}}(\pi' \| \pi)(s) \tag{81}$$

where $D_{\mathrm{TV}}$ is the Total Variation Distance. Furthermore, by Pinsker's inequality, we have that

$$D_{\mathrm{TV}}(\pi' \| \pi) \leq \sqrt{\frac{1}{2} D_{\mathrm{KL}}(\pi' \| \pi)}, \tag{82}$$

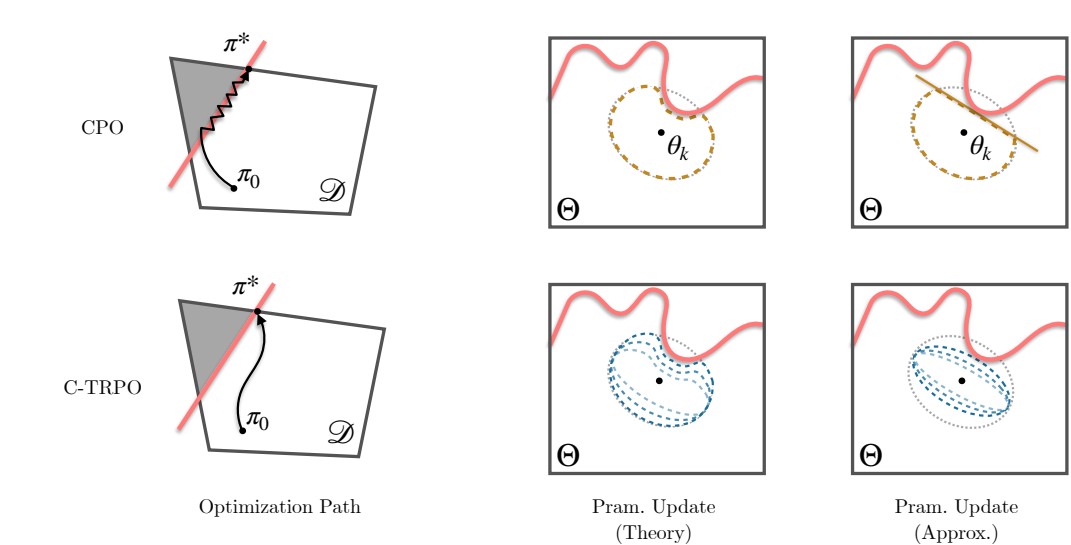

Figure 6: Pictorial illustration of conceptual and practical differences between CPO and C-TRPO. The local approximation of C-TRPO's trust region results in a single quadratic constraint, which is compressed in the direction of the closest cost surface, depending on the hyper-parameter $\beta$ (blue dashed lines on the right). This is in contrast to CPO, where the local approximation of the update results in a quadratic constraint which is not affected by the cost, and a linear constraint which only takes effect upon contact with the cost surface. Intuitively, this results in a smoother optimization path for C-TRPO that remains on the interior of the safe policy space for longer.

and by Jensen's inequality

$$\mathbb{E}_{s \sim d_\pi} D_{\text{TV}}(\pi' || \pi)(s) \leq \sqrt{\frac{1}{2} \mathbb{E}_{s \sim d_\pi} D_{\text{KL}}(\pi' || \pi)(s)}, \tag{83}$$

It follows that we can not only substitute the KL-divergence into the bound but any divergence

$$D_\Phi(d'_\pi || d_\pi) \geq \mathbb{E}_{s \sim d_\pi} D_{\text{KL}}(\pi' || \pi)(s) \tag{84}$$

can be substituted, and still retains TRPO's and CPO's update guarantees.

### C.4 COMPARISON WITH CPO

In the approximate case of C-TRPO and CPO, where the reward is approximated linearly, and the trust region quadratically, the constraints differ in that C-TRPO's constraint is

$$(\theta - \theta_k)(\bar{H}_{\text{KL}}(\theta) + \beta \bar{H}_\phi(\theta))(\theta - \theta_k) < \delta$$

whereas CPO's is

$$(\theta - \theta_k)\bar{H}_{\text{KL}}(\theta)(\theta - \theta_k) < \delta \text{ and } V_c^{\theta_k} + (\nabla_\theta \mathbb{A}_c^{\theta_k}(\theta))^\top (\theta - \theta_k) \leq b.$$

Figure 6 illustrates the differences between CPO and C-TRPO.

## D ADDITIONAL EXPERIMENTS

### D.1 EFFECTS OF THE HYPER-PARAMETERS

To better understand the effects of the two hyperparameters $\beta$ and $b_{\text{H}}$, we observe how they change the training dynamics through the example of the *AntVelocity* environment.

The safety parameter $\beta$ modulates the stringency with which C-TRPO satisfies the constraint, without limiting the expected return for values up to $\beta = 1$, see Figure 7. For higher values, the expected

return starts to degrade, partly due to $\bar{D}_\phi$ being relatively noisy compared to $\bar{D}_{\text{KL}}$ and thus we recommend the choice $\beta = 1$.

Further, we observe that constraint satisfaction is stable across different choices of cost threshold $b$, see Figure 8, and that in most environments, constraint violations seem to reduce as the algorithm converges, meaning that the regret flattens over time. This behavior suggests that the divergence estimation becomes increasingly accurate over time, potentially allowing C-TRPO to achieve sublinear regret. However, we leave regret analysis of the finite sample regime for future research.

Finally, employing a hysteresis fraction $0 < b_{\text{H}} < b$ seems beneficial, possible because it leads the iterate away from the boundary of the safe set, and because divergence estimates tend to be more reliable for strictly safe policies. The effect of the choice of $b_{\text{H}}$ is visualized in Figure 10.

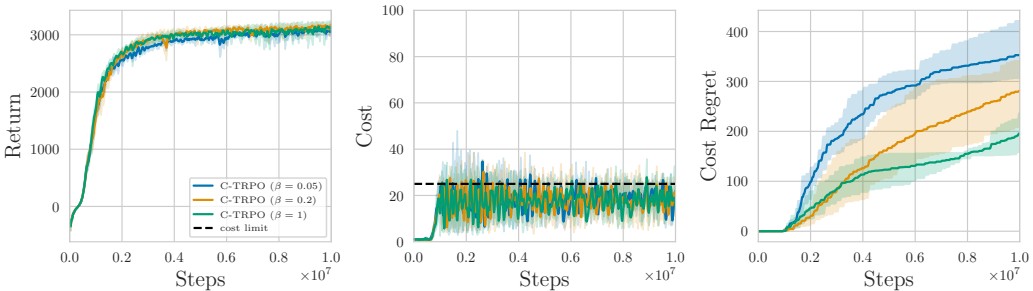

Figure 7: Changing $\beta$ influences the degree of safety.

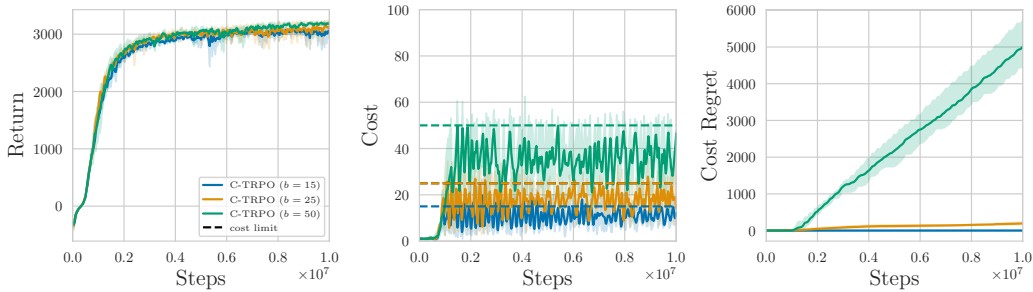

Figure 8: The constraint satisfaction is robust to changing the cost limit.

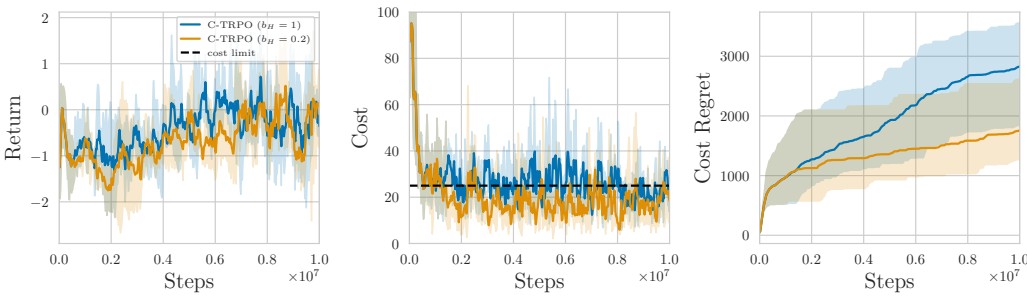

Figure 9: In difficult environments, e.g. those that start off in the unsafe policy set, it seems to be beneficial to set a fraction of the cost limit for hysteresis.

## D.2 ABLATION STUDY: CPO VS. C-TRPO

We conduct an ablation study to rule out that our improvements of C-TRPO over CPO are only due to hysteresis. For this, we run both CPO and C-TRPO with and without hysteresis with the same hysteresis parameter as in our other experiments. We see that the hysteresis improves safety for both algorithm. Further, we find that the hysteresis slightly reduces the return of C-TRPO. Overall, we clearly see that C-TRPO itself is much safer compared to CPO as even C-TRPO without hysteresis achieves lower cost regret compared to CPO with hysteresis.

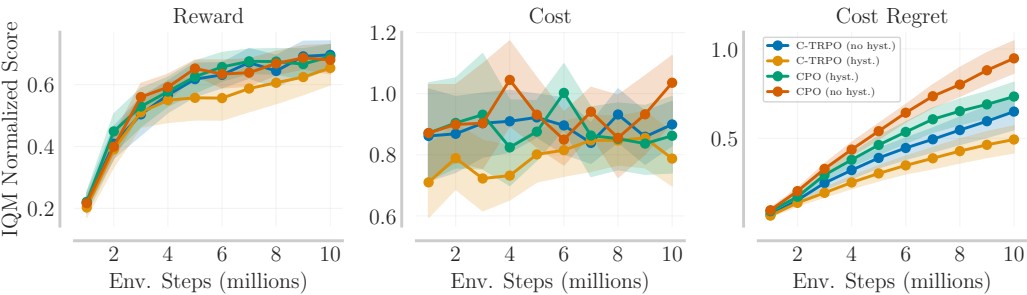

Figure 10: Ablation study on the core components of C-TRPO: Safe trust region (C-TRPO no hyst.) and recovery with hysteresis (CPO hyst.). Evaluation is based on the Inter Quartile Mean (IQM) normalized scores across 5 seeds and 8 tasks. From left to right: episode return of the reward (PPO normalized), episode return of the cost (threshold normalized), and cumulative cost violation (CPO normalized).

## D.3 PERFORMANCE ON INDIVIDUAL ENVIRONMENTS

Here, we compare C-TRPO to CPO and PCPO as representative baselines on all individual environments in terms of their sample efficiency curves.

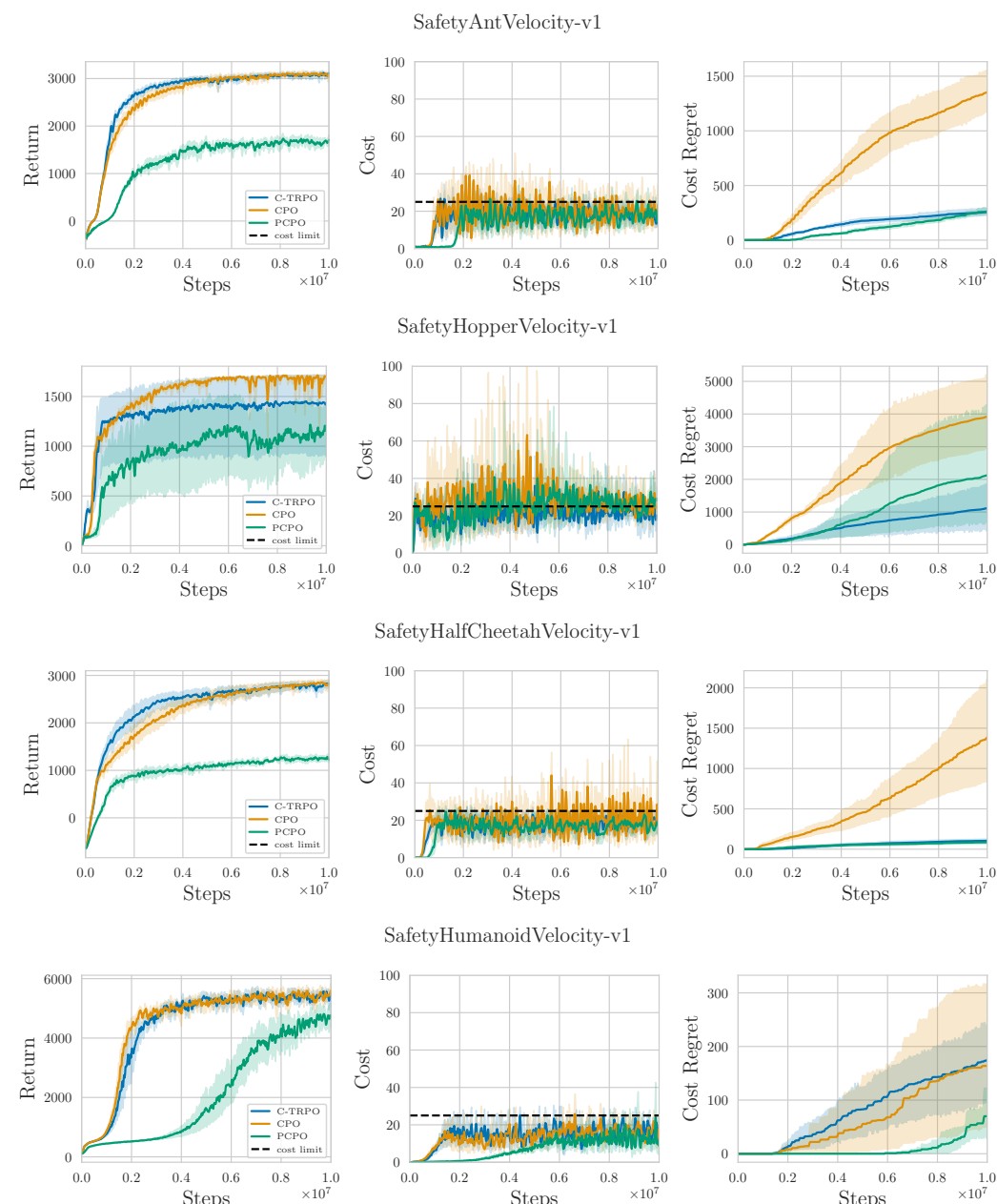

Figure 11: C-TRPO vs. CPO and PCPO in the locomotion environments.

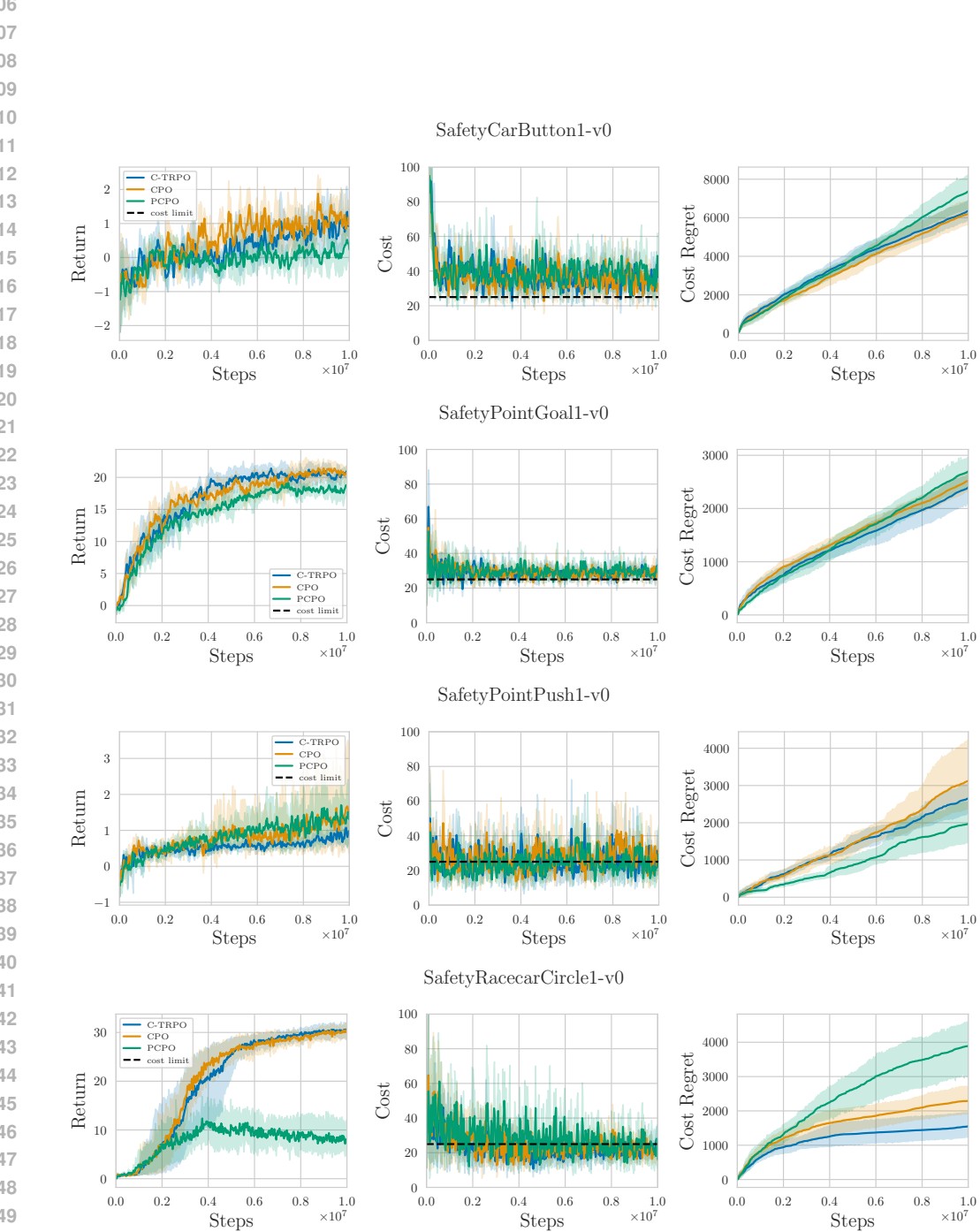

Figure 12: C-TRPO vs. CPO and PCPO in the navigation environments.

Table 1: Average evaluation performance per task across 10 evaluation runs and 5 seeds each. We highlight the best performance with respect to the average return $V_r$ in bold, and underline the lowest average cost $V_c$. Note that the table only contains information about the final evaluation performance, not about cost violations during training.

| | C-TRPO | | C-TRPO-HYST | | CPO | | CPO-HYST | |
|---|---|---|---|---|---|---|---|---|
| | $V_r$ | $V_c$ | $V_r$ | $V_c$ | $V_r$ | $V_c$ | $V_r$ | $V_c$ |
| **AntVelocity** | 2810.1 ± 45.2 | 7.9 ± 7.6 | 2786.5 ± 75.8 | 9.0 ± 8.2 | 2569.7 ± 61.4 | 5.7 ± 4.9 | 2629.3 ± 142.9 | 16.8 ± 30.6 |
| **HalfCheetahVelocity** | 2316.6 ± 223.9 | 8.9 ± 6.9 | 2340.8 ± 220.3 | 14.6 ± 10.1 | 1990.1 ± 191.2 | 20.8 ± 19.9 | 1921.5 ± 230.5 | 13.4 ± 11.0 |
| **HumanoidVelocity** | 4837.7 ± 745.7 | 3.1 ± 3.9 | 5367.1 ± 292.3 | 11.0 ± 19.0 | 5654.3 ± 67.5 | 0.0 ± 0.0 | 5583.4 ± 124.3 | 0.5 ± 1.0 |
| **HopperVelocity** | 1361.2 ± 463.7 | 12.6 ± 12.5 | 1358.8 ± 469.6 | 13.6 ± 15.0 | 1432.0 ± 40.5 | 0.4 ± 0.8 | 1416.8 ± 99.5 | 3.0 ± 3.0 |
| **CarButton1** | -1.6 ± 1.9 | 59.1 ± 29.2 | -1.1 ± 1.9 | 45.0 ± 6.7 | -1.4 ± 1.1 | 78.3 ± 54.5 | -3.4 ± 4.3 | 38.3 ± 34.3 |
| **PointGoal1** | 13.3 ± 4.2 | 32.9 ± 6.0 | 10.0 ± 2.0 | 23.7 ± 9.4 | 12.6 ± 2.9 | 22.5 ± 5.9 | 15.3 ± 6.4 | 18.0 ± 14.2 |
| **RacecarCircle1** | 8.3 ± 7.0 | 35.1 ± 25.4 | 6.9 ± 7.2 | 15.9 ± 15.8 | 8.3 ± 7.9 | 35.6 ± 14.5 | 9.1 ± 6.0 | 27.6 ± 19.6 |
| **PointPush1** | 0.0 ± 0.5 | 21.5 ± 14.2 | **0.8 ± 0.6** | 17.1 ± 18.3 | 0.3 ± 0.3 | 71.7 ± 59.8 | **0.8 ± 0.4** | 9.6 ± 11.0 |

| | PCPO | | FOCOPS | | CUP | | P3O | |
|---|---|---|---|---|---|---|---|---|
| | $V_r$ | $V_c$ | $V_r$ | $V_c$ | $V_r$ | $V_c$ | $V_r$ | $V_c$ |
| **AntVelocity** | 2064.1 ± 119.0 | 53.3 ± 47.1 | 2374.1 ± 249.5 | 194.5 ± 50.6 | 1853.7 ± 322.1 | 26.7 ± 33.8 | 1475.5 ± 160.2 | 2.8 ± 3.7 |
| **HalfCheetahVelocity** | 1424.5 ± 130.4 | 66.3 ± 11.6 | 2216.0 ± 137.7 | 6.2 ± 10.9 | **2511.0 ± 146.8** | 35.1 ± 64.0 | 2120.0 ± 218.1 | 7.9 ± 12.6 |
| **HumanoidVelocity** | 585.2 ± 27.7 | 0.0 ± 0.0 | 1304.4 ± 681.6 | 16.8 ± 23.5 | 1406.4 ± 403.5 | 3.0 ± 2.5 | 709.1 ± 181.2 | 0.6 ± 0.7 |
| **HopperVelocity** | 798.4 ± 407.9 | 11.7 ± 13.1 | 1478.3 ± 105.8 | 22.2 ± 44.0 | **1538.4 ± 83.7** | 44.3 ± 72.6 | 1504.6 ± 98.4 | 4.0 ± 7.9 |
| **CarButton1** | -2.0 ± 3.2 | 81.7 ± 43.2 | -6.9 ± 6.8 | 26.1 ± 26.6 | 1.3 ± 2.9 | 60.1 ± 67.3 | -0.6 ± 0.6 | 39.1 ± 29.0 |
| **PointGoal1** | 12.2 ± 2.4 | 28.7 ± 10.3 | 17.8 ± 3.9 | 53.0 ± 23.9 | 17.6 ± 7.4 | 39.7 ± 17.7 | 3.1 ± 1.2 | 32.6 ± 17.8 |
| **RacecarCircle1** | 6.7 ± 5.2 | 22.1 ± 16.3 | 5.6 ± 5.0 | 14.8 ± 27.8 | 17.1 ± 6.2 | 26.6 ± 22.6 | 2.1 ± 1.0 | 52.6 ± 36.4 |
| **PointPush1** | 0.4 ± 0.5 | 26.8 ± 41.3 | 0.3 ± 0.4 | 33.2 ± 51.1 | 0.4 ± 0.2 | 12.9 ± 10.7 | 0.2 ± 0.4 | 4.7 ± 6.2 |

| | IPO | | CPPO-PID | | TRPO-LAG | | PPO-LAG | |
|---|---|---|---|---|---|---|---|---|
| | $V_r$ | $V_c$ | $V_r$ | $V_c$ | $V_r$ | $V_c$ | $V_r$ | $V_c$ |
| **AntVelocity** | 1690.4 ± 322.3 | 7.5 ± 7.1 | 1793.2 ± 248.0 | 18.7 ± 22.9 | **2894.4 ± 124.8** | 14.6 ± 8.0 | 1840.1 ± 263.7 | 19.7 ± 24.9 |
| **HalfCheetahVelocity** | 2053.3 ± 204.8 | 36.1 ± 57.5 | 2338.2 ± 196.9 | 6.4 ± 8.1 | 2449.4 ± 213.6 | 14.6 ± 12.0 | 2360.2 ± 209.0 | 2.8 ± 5.1 |
| **HumanoidVelocity** | 2685.0 ± 1357.3 | 10.6 ± 9.4 | 4280.2 ± 1288.6 | 4.3 ± 3.6 | **5696.6 ± 90.5** | 0.0 ± 0.0 | 4192.4 ± 1108.5 | 6.4 ± 6.2 |
| **HopperVelocity** | 1224.0 ± 424.0 | 4.6 ± 6.0 | 1490.7 ± 121.0 | 2.8 ± 5.5 | 500.4 ± 434.5 | 19.6 ± 15.1 | 100.3 ± 26.9 | 4.2 ± 7.5 |
| **CarButton1** | -0.3 ± 1.0 | 31.1 ± 17.3 | -2.0 ± 1.8 | 18.6 ± 7.7 | -9.4 ± 5.9 | 29.4 ± 21.2 | **2.4 ± 1.0** | 113.8 ± 53.1 |
| **PointGoal1** | 2.0 ± 1.1 | 30.8 ± 13.6 | 1.6 ± 2.0 | 46.3 ± 39.3 | **25.0 ± 0.5** | 44.6 ± 6.8 | 18.9 ± 2.2 | 49.7 ± 20.1 |
| **RacecarCircle1** | 0.9 ± 0.1 | 42.0 ± 30.2 | 1.0 ± 0.2 | 35.4 ± 31.5 | **24.8 ± 3.3** | 5.6 ± 2.4 | 9.7 ± 4.1 | 3.6 ± 2.5 |
| **PointPush1** | 0.4 ± 0.6 | 25.0 ± 32.1 | 0.2 ± 0.2 | 30.5 ± 16.6 | 0.6 ± 0.6 | 2.1 ± 1.9 | 0.4 ± 0.3 | 24.9 ± 12.1 |

