# OpenReview forum: "Embedding Safety into RL: A New Take on Trust Region Methods"
_ICLR.cc/2025/Conference — Submitted to ICLR 2025_

### Official Review · Reviewer_jupc · 2024-10-24

**Soundness:** 3
**Presentation:** 3
**Contribution:** 3
**Rating:** 8
**Confidence:** 3

**Summary:**

This paper introduces a novel policy optimization method for safe RL by constructing trust region of each iteration within the safe policy set for update.

**Strengths:**

By constructing trust region within the safe policies set, this method maintains competitive returns with less constraint violations during training. Since only construction of trust region is altered, this method still preserve convergence and policy improvement guarantee of original TRPO. This paper is technically solid and well-written. In the analysis part, author also provides a thorough explanation on connection between C-TRPO and CPO on policy update and constraint violation.

**Weaknesses:**

1, line 142 CMPD &rarr; CMDP\
2, line 354 Proposition 1 refers to Theorem 1?

**Questions:**

1, For safety during training and convergence of constrained natural policy gradient, what kind of initial set assumptions are needed?
2, It would be interesting to see some comparison with hard constraints based approaches such as control barrier function based method[1, 2, 3], since similar notion of invariance seems to be brought up in section 4.2 to ensure safety during training.


[1]Charles Dawson, Sicun Gao, and Chuchu Fan. Safe control with learned certificates: A survey of neural lyapunov, barrier, and contraction methods for robotics and control. IEEE Transactions on Robotics, 2023.\
[2]Yixuan Wang, Simon Sinong Zhan, Ruochen Jiao, Zhilu Wang, Wanxin Jin, Zhuoran Yang, Zhaoran Wang, Chao Huang, and Qi Zhu. Enforcing hard constraints with soft barriers: Safe reinforcement learning in unknown stochastic environments. In International Conference on Machine Learning, pages 36593–36604. PMLR, 2023b.
[3]Jason Choi, Fernando Castaneda, Claire J Tomlin, and Koushil Sreenath. Reinforcement learning for safety-critical control under model uncertainty, using control lyapunov functions and control barrier functions. arXiv preprint arXiv:2004.07584, 2020.

---

> ### Author Response · Authors · 2024-11-23
> **Response to Reviewer jupc**
>
> # Weaknesses
>
> > - line 142 CMPD → CMDP
> > - line 354 Proposition 1 refers to Theorem 1?
>
> Thank you for catching those typos, which we have corrected.
>
> # Questions
>
> > For safety during training and convergence of constrained natural policy gradient, what kind of initial set assumptions are needed?
>
> Theorem 5 and Theorem 6, which guarantees safety during training and convergence only assume that the initial policy $\pi_0$ strictly satisfies the safety constraints and that $\phi'(x)\to+\infty$ for $x\searrow 0$. Please let us know, if there are any follow up questionson this or if you have any other suggestions that can help prevent missunderstanding.
>
> > It would be interesting to see some comparison with hard constraints based approaches such as control barrier function based method[1, 2, 3], since similar notion of invariance seems to be brought up in section 4.2 to ensure safety during training.
>
> We agree that connections to control barrier functions would be interesting to explore, and think that this is a promising avenue for further research.
>
> On the surface, our approach seems quite different, since barrier function methods are usually applied to deterministic dynamical systems with hard constraints. However, there seems to be a connection when we view the policy parameter space as the state space of a dynamical system.
>
> Our current understanding is that, in the language of [4, 5], the safe occupancy set is *foward invariant* under the dynamics of the C-NPG flow, with *control barrier function*
> $$
>     h(d) = \sum_i b_i-\sum_{s,a} c_i(s,a) d_\pi(s,a),
> $$
> where $d$ is the occupancy measure, and *reciprocal barrier function*
> $$
>     B(d) = \phi(h(d)),
> $$
> where $\phi$ is a logarithmic barrier function.
>
> Hence, our extra term in the safe mirror function can be seen as a reciprocal barrier function in the framework of control barrier functions.
>
> Interestingly, the authors of [5] report that their approach is "Motivated by the barrier method in optimization", which is also our motivation for C-TRPO and C-NPG. However, the central difference is that [5] considers barrier functions as "test" functions, where we consider the Bregman divergence and hence the convex geometry induced by such barrier functions.
>
> ### Conclusion
>
> Thanks for this interesting discussion! Please let us know if you have follow-up questions.
>
> [4] Ames, Aaron D., et al. "Control barrier functions: Theory and applications." 2019 18th European control conference (ECC). IEEE, 2019.
> [5] Ames, Aaron D., et al. "Control barrier function based quadratic programs for safety critical systems." IEEE Transactions on Automatic Control 62.8 (2016): 3861-3876.

---

### Official Review · Reviewer_C9Zt · 2024-10-28

**Soundness:** 3
**Presentation:** 3
**Contribution:** 2
**Rating:** 5
**Confidence:** 2

**Summary:**

This work proposes Constrained Trust Region Policy Optimization (C-TRPO) that aims to ensure safe exploration (always being safe during the training) without sacrificing performance in reward. Inspired by TRPO, the main idea of this work is to incorporate cost constraints into divergence to create safer trust regions. The divergence is obtained by mapping the policy onto its occupancy measure on the state-action polytope, where a safe geometry can be defined using standard tools from convex optimization.

**Strengths:**

Overall, this paper is well-written and well-presented.

The authors honestly point out and discuss the similarities and differences from the existing literature, and cite the paper correctly.

Some figures in the paper are intuitive such as Figure 2.

Overall, the mathematical proofs are sound.

I indeed have several concerns regarding this work, and I hope some of them can be answered or addressed after rebuttal.

**Weaknesses:**

Line 43. Does " without sacrificing performance" mean C-TRPO can achieve exactly the same performance as that of TRPO (unconstrained RL)? The experiment does not support this. Indeed, Figure 3 shows that C-TRPO is even a bit worse than CPO. (TRPO should be have even much higher return as it is unconstrained.)


Line 60-62, please provide more details that why model-based safe RL is less general, and what kind of stricter guarantees they provided.


Line 74-75. I am not sure if I argree with this. In the convex problems (which I understand may not hold in RL) and the problems where the policy parameterization is "perfect", there is no "bias". Solving the langragian-weighted objective is as good as solving the constrained RL. See the reference below

"Paternain, S., Chamon, L., Calvo-Fullana, M., & Ribeiro, A. (2019). Constrained reinforcement learning has zero duality gap. Advances in Neural Information Processing Systems, 32."


Line 76-82. The dicussion of trust region methods is too short. It is even shorter than the Penalty methods, while the paper focuses on the trust region methods.


Line 86-87. I don't understand. C-TRPO is an approximation of C-TRPO itself?


Line 112-120. Please make it clear in the formula that the expectation is w.r.t. initial state distribution, policy, and the transition function. "the expectations are taken over trajectorie" is too brief and not clear.


Line 188-189. If I remember correctly, doesn't CPO already inherit TRPO's update guarantees for reward and constraints?


Line 188. Refer to Figure 1 too early. There is no enough explaination in the main text or the caption of Figure, e.g., what is \beta, etc.

From Figure 1, why the proposed method is better than CPO? One is a clipped policy space, and the other one is a newly constructed policy space. It is hard to see which one is better intuitively. It also seems like C-TRPO has the same bounds as that of CPO (on page 7). The novelty is a bit limited in this sense.

To be honest, it is hard to tell if C-TRPO is better than baselines from Figure 3. Especially that it has lower return than CPO.

In general, I am a bit worried about the novelty of this work. It seems to me that there is not too much change compared to TRPO and CPO. Especially that Figure 1 does not clearly explain the difference. Why the fourth is better? Also, are these figures just hand-drawn intuition illustration? Are they true in practice?

Enhance the writing and fix typos, e.g., Line 63, Line 142,

**Questions:**

Please see my questions in the "Weaknesses" section.

---

> ### Author Response · Authors · 2024-11-23
> **Response to Reviewer C9Zt (pt 1/2)**
>
> # Weaknesses:
>
> > Line 43. Does " without sacrificing performance" mean C-TRPO can achieve exactly the same performance as that of TRPO (unconstrained RL)? The experiment does not support this. Indeed, Figure 3 shows that C-TRPO is even a bit worse than CPO. (TRPO should be have even much higher return as it is unconstrained.)
>
> We did not mean to claim that constraint satisfaction doesn't come with performance dagradation compared to the unconstrained TRPO. Instead, C-TRPO noticably improves upon the *amount of cost violation during training* compared to existing CMDP algorithms (like CPO) without any reduction in exected reward, i.e. "without sacrificing performance". We updated the pdf in lines 43 to make it clear that we compare C-TRPO with other safe optimizers and not TRPO.
>
> > Line 60-62, please provide more details that why model-based safe RL is less general, and what kind of stricter guarantees they provided.
>
> Here, we refer to approaches like [1]. We have changed our wording here, please let us know, if this does not resolve your concerns.
>
> [1] Berkenkamp, Felix, et al. "Safe model-based reinforcement learning with stability guarantees." Advances in neural information processing systems 30 (2017).
>
> > Line 74-75. I am not sure if I argree with this. In the convex problems (which I understand may not hold in RL) and the problems where the policy parameterization is "perfect", there is no "bias". Solving the langragian-weighted objective is as good as solving the constrained RL. See the reference below
> > "Paternain, S., Chamon, L., Calvo-Fullana, M., & Ribeiro, A. (2019). Constrained reinforcement learning has zero duality gap. Advances in Neural Information Processing Systems, 32."
>
> If the penalty coefficient is a fixed hyperparameter, then the objective function will be changed, and there will potentially be a gap between the optimal solution of the constrained problem and the unconstrained problem with penalty (see e.g. Theorem 1. in the IPO paper).
> We agree that this may not be true in general for Lagrangian approaches.
>
> We have expanded the related work section and now discuss Lagrangian and penalty methods in separate paragraphs.
>
> > Line 76-82. The dicussion of trust region methods is too short. It is even shorter than the Penalty methods, while the paper focuses on the trust region methods.
>
> We agree with the criticism and have expanded our discussion of trust region methods in the related work section. We also discuss their relation to our proposed method in more detail.
>
> > Line 86-87. I don't understand. C-TRPO is an approximation of C-TRPO itself?
>
> Our wording comes from the analogy with the original TRPO algorithm, where the authors introduce a theoretical TRPO update, which is intractable in deep RL scenarios, and an approximation using the policy advantage.
>
> Both are referred to as TRPO in the literature. However, we agree that this can be confusing, and have changed the main text to exclusively refer to the practical algorithm as C-TRPO. We have updated the PDF accordingly and improved the writing to reflect this.
>
> > Line 112-120. Please make it clear in the formula that the expectation is w.r.t. initial state distribution, policy, and the transition function. "the expectations are taken over trajectorie" is too brief and not clear.
>
> Thank you for your suggestions, which we have followed.
>
> > Line 188-189. If I remember correctly, doesn't CPO already inherit TRPO's update guarantees for reward and constraints?
>
> You are correct. Here we point out why trust region methods are attractive despite the recent focus on penalty methods: "The main advantage of formulating a safe policy optimization algorithm as a trust region
> method is that it inherits TRPO’s update guarantees for both the reward as well as the constraints" This applies equally to CPO and motivates why we want to improve it in C-TRPO.
>
> We move this entire discussion to the introduction after the related work section on trust region methods.
>
> > Line 188. Refer to Figure 1 too early. There is no enough explaination in the main text or the caption of Figure, e.g., what is \beta, etc.
>
> Thank you for catching this. We change the caption of Figure 1 and its reference to better fit into the flow of the main text.

---

> ### Author Response · Authors · 2024-11-23
> **Response to Reviewer C9Zt (pt 2/2)**
>
> > From Figure 1, why the proposed method is better than CPO? One is a clipped policy space, and the other one is a newly constructed policy space. It is hard to see which one is better intuitively. It also seems like C-TRPO has the same bounds as that of CPO (on page 7). The novelty is a bit limited in this sense.
>
> In Constrained Policy Optimization (CPO), trust regions are intersected with the set of safe policies, meaning that updates only differ from those in Trust Region Policy Optimization (TRPO) when TRPO would leave the safe region. However, this approach allows updates to lie on the constraint surface, which can increase the probability of constraint violations due to estimation errors. In contrast, Constrained TRPO (C-TRPO) adapts the trust regions so that they are always contained within the safe region, even when far from the constraint surface. The design of geometries depending on the constraint set is much more natural from a convex optimization point of view than incorporating them as hard constraints in a trust region method, see for example Alvarez et al. (2004). Figure 1 shows the modification of the trust regions for C-TRPO and different values of $\beta$. We see that with increasing values of $\beta$ the trust regions become flatter. This modification encourages updates of C-TRPO to move more parallel to the constraint surfaces rather than directly toward them, as illustrated in Figure 2. As a result, C-TRPO reduces the likelihood of constraint violations during training compared to CPO. To clarify this distinction and the motivation behind safe trust regions, we have added a dedicated paragraph in the introduction, which we hope provides a clearer explanation.
>
> It is true that the performance bounds for CPO are analogous to the ones for C-TRPO. However, it is important to note that the guarantees for the constraint violations are tighter for C-TRPO. The reason is that the trust regions of C-TRPO incorporate the constraints, which allows for a tighter control of the cost after the update. See also the discussion below Proposition 3.
>
> > To be honest, it is hard to tell if C-TRPO is better than baselines from Figure 3. Especially that it has lower return than CPO.
>
> Statistically, C-TRPO has indistinguishable return from CPO, but has significantly less constraint violations, as can be seen on the right hand column of Figure 3. Conversely, the only methods that achieve comparable constraint violation, which are PCPO and P3O, achieve much worse return. Hence, C-TRPO offers comptetitive return with improved safety during training with minimal to no overhead. Of course, it depends on the specific application, whether constraint violation during training is relevant.
>
> > In general, I am a bit worried about the novelty of this work. It seems to me that there is not too much change compared to TRPO and CPO. Especially that Figure 1 does not clearly explain the difference. Why the fourth is better? Also, are these figures just hand-drawn intuition illustration? Are they true in practice?
>
> We see that Figure 1, in particular the label *C-TRPO/CPO ($\beta=10^{-4}$)* could cause some confusion. We meant to convey that C-TRPO with small $\beta$ approximates CPO. We have updated the figure to prevent the missconception that CPO is an actual special case of C-TRPO. CPO uses a hard constraint, whereas C-TRPO encodes the constraints into the divergence function and thus the trust region, which feels more natural from a convex optimization perspective. The updates of CPO only differ from those of TRPO when TRPO would leave the safe region and allows for updates that lie at the constraint surface. In comparison, C-TRPO uses trust regions that are flatter parallel to the constrained surface. This naturally encourages updates to move more parallel to the constraint surfaces and thereby reduces the likelihood of constraint violations during training compared to CPO. To conclude, we see C-TRPO as a natural extension of TRPO to constrained MDPs and as a new way of incorporating the safety constraints. As such we kindly disagree that C-TRPO is essentially the same as TRPO and CPO and hope that our explanation helps understanding the differences. If there are any other ways how we can make this more clear, please do not hesitate to let us know.
>
> > Enhance the writing and fix typos, e.g., Line 63, Line 142,
>
> Thanks for catching those. We update the manuscript accordingly.
>
> ### Conclusion
>
> Thank you for your thorough evaluation! We hope our responses address your comments. Please let us know if you have follow-up questions. We are happy to discuss.

---

> > ### Comment · Reviewer_C9Zt · 2024-11-26
> >
> > Thank you for your detailed reply, which indeed addressed part of my concerns. I will increase my score to 4 for now, and will also pay attention to the authors' discussion with other reviewers.

---

> > > ### Author Response · Authors · 2024-11-26
> > >
> > > Thank you for your reply. Please feel free to let us know, which of your questions and concerns remain, so that we can try to address them.
> > >
> > > Best wishes

---

### Official Review · Reviewer_ujBB · 2024-10-31

**Soundness:** 2
**Presentation:** 3
**Contribution:** 2
**Rating:** 3
**Confidence:** 4

**Summary:**

This paper presents a constrained optimization method called Constrained Trust Region Policy Optimization (C-TRPO), which modifies the geometry of the policy space based on safety constraints to create trust regions comprised solely of safe policies. They also provide an approximate implementation of C-TRPO. The main contribution is integrating safety constraints into policy divergence without introducing additional computational complexity or bias. The theoretical analysis of convergence is also provided. Experimental results show that C-TRPO achieves comparable policy performance and smaller constraint violations compared to common safe optimization methods.

**Strengths:**

## Originality and Significance

1) Safe RL is a crucial direction in reinforcement learning, which has significant implications for the application of reinforcement learning in real-world scenarios.

2) This paper proposed the approach C-TRPO to address the constrained optimization problem by modifying policy divergence, which appears to be novel.

## Quality and Clarity
1) This paper provides mathematical formulations for the main concepts needed to understand the approach. They also provide relevant theoretical results.

2) The paper includes  a number of Figures which are helpful in understanding the main concepts in the paper. The use of figures (such as Figure 1 to illustrate the constrained KL - divergence in policy space) and examples (like the description of the optimization trajectories in Figure 2) enhances the clarity of the explanations.

3) The optimization implementation and approximation process is provided in detail.

**Weaknesses:**

1) The motivation and impact of integrating safety constraints into policy divergence are not sufficiently clear.

2) The core idea of this paper is to incorporate the constraints into the policy divergence, but according to the definition in equation (15), the divergence approaches $\infty$ when the policy approaches the constraint boundary, which results in the new divergence $D_c$ failing to satisfy the constraints, potentially leading to the absence of a solution.

3) The paper does not provide sufficient evidence to prove that the improved effectiveness of C-TRPO is solely due to the new policy divergence. It states that the enhanced constraint satisfaction compared to CPO is attributed to a slowdown and reduction in the frequency of oscillations around the cost threshold. This effect may also be partially due to the hysteresis-based recovery mechanism. However, the paper does not demonstrate whether introducing the same hysteresis-based recovery mechanism to CPOs would yield similar improvements.

4) Some of the theoretical explanations in the paper are not clear.

## Experiments
1) The paper does not include state-of-the-art baselines. It would be beneficial to compare C-TRPO with some of the latest safe RL algorithms to verify its effectiveness.

2) No ablation studies have been conducted to assess the roles of the core components in C-TRPO.

3) The observed results improvement is limited. The experimental results in the appendix indicate that the constraints in C-TRPO appear to be at the same level as in CPO, showing no smaller constraint violations (e.g., in safetycarbutton1 and safetyracecarcircle1).

4) No code is provided, raising concerns about reproducibility.

**Questions:**

1) In Proposition 3, when $\beta = 0$, $D_C = D_{KL}$ according to equation (9), why does C-TRPO approach CPO but not TRPO in this case？

2) In Proposition 4, according to the proof in the appendix,  $\mathbb{A}_c < \Psi^{-1}$, Why is the upper bound of C-TRPO smaller than that of CPO? Could the authors provide a more detailed explanation of this upper bound?

3) As the policy approaches the constraint boundary, $D_\phi$ in equation 15 will approach infinity, which may make Equation (14) unsatisfiable and results in no solution. How is this situation addressed in the proposed framework?

I am willing to raise my score if the authors can address my concerns.

---

> ### Author Response · Authors · 2024-11-23
> **Response to Reviewer ujBB (pt 1/2)**
>
> # Weaknesses
> > The motivation and impact of integrating safety constraints into policy divergence are not sufficiently clear.
>
> We expanded the introduction and section 3 to improve exposition. We hope that the motivation for our approach is more clear now, and kindly refer to the general reply for a brief summary of the main points.
>
> > The core idea of this paper is to incorporate the constraints into the policy divergence, but according to the definition in equation (15), the divergence approaches when the policy approaches the constraint boundary, which results in the new divergence failing to satisfy the constraints, potentially leading to the absence of a solution.
>
> See questions below.
>
> > The paper does not provide sufficient evidence to prove that the improved effectiveness of C-TRPO is solely due to the new policy divergence. It states that the enhanced constraint satisfaction compared to CPO is attributed to a slowdown and reduction in the frequency of oscillations around the cost threshold. This effect may also be partially due to the hysteresis-based recovery mechanism. However, the paper does not demonstrate whether introducing the same hysteresis-based recovery mechanism to CPOs would yield similar improvements.
>
> See experiments below.
>
> > Some of the theoretical explanations in the paper are not clear.
>
> We re-wrote the CPO equivalence proof and add some intution about why this is true. We hope that this improves the readability of the theoretical parts of the paper. Please let us know if there are other theoretical results, for which we can improve exposition.
>
> ## Experiments
> > The paper does not include state-of-the-art baselines. It would be beneficial to compare C-TRPO with some of the latest safe RL algorithms to verify its effectiveness.
>
> We agree that comparison to recent baselines is beneficial and further include FOCOPS and CUP as more recent trust region methods, and IPO and P3O as (barrier) penalty baselines.
>
> > No ablation studies have been conducted to assess the roles of the core components in C-TRPO.
>
> We agree with the criticism and include an ablation study involving CPO and C-TRPO with and without hysteresis.
>
> > The observed results improvement is limited. The experimental results in the appendix indicate that the constraints in C-TRPO appear to be at the same level as in CPO, showing no smaller constraint violations (e.g., in safetycarbutton1 and safetyracecarcircle1).
>
> It is true, that constraint violations are similar for CPO and C-TRPO in some environments. We believe that this is due to stochasticity in the algorithms and in some of the environments, and may also be due to the specific geometries of the optimization problems for some of them. In future research, we want to better understand where exactly C-TRPO improvements have minor impact and why.
>
> However, across multiple environments, C-TRPO clearly outperforms CPO in constraint violation, as can be seen in column 3 of figure 3, and has similar final return, especially due to the superior performance in the locomotion tasks.
>
> While running the ablations, we also noticed a small mistake in our environment-wise evaluations. We had run C-TRPO with $\beta=0.1$ as opposed to our recommendation $\beta=1.0$. We updated the environment-wise plots to show C-TRPO ($\beta=1.0$), which is better in terms of constraint satisfaction on some environments, for example SafetyRacecarCircle.
>
> >  No code is provided, raising concerns about reproducibility.
>
> We did not intend to publish the repository at this point in time, but to avoid concerns about reproducibility, we offer an anonymized repo: https://anonymous.4open.science/r/c-trpo/
>
> Please bear in mind, that the repository is still in a rough shape due to time constraints on our end. But we intend to clean it up before publication.

---

> ### Author Response · Authors · 2024-11-23
> **Response to Reviewer ujBB (pt 2/2)**
>
> # Questions
> > In Proposition 3, when, according to equation (9), why does C-TRPO approach CPO but not TRPO in this case？
>
> We hope that the following explanation provides some additional intuition.
>
> **Edit:** The reviewer has rightfully pointed out typos in this response, which we have edited. These are 1) a missing $\beta$ in $D_C$, i.e. $\bar D_C = \bar D_{KL} + \bar D_\phi \rightarrow \bar D_C = \bar D_{KL} + \beta \bar D_\phi$, and 2) the inequality in the definition of safe policies $V_c(\pi_k) < b \rightarrow V_c(\pi_k)\leq b$. Otherwise, the answer still applies.
>
> Recall that $\bar D_C = \bar D_{KL} + \beta \bar D_\phi$. In C-TRPO's divergence, $\beta \bar D_\phi$ is a function of the cost advantage $\mathbb{A}_c$, and as $\beta$ approaches 0, it approaches a "step" or "indicator" function
>
> $$
> \lim_{\beta\to+0} \beta D_\phi(\mathbb{A}_c) = \mathbb{I}(\{\mathbb{A}_c < b-V_c(\pi)\}),
> $$
> which is 0 if $\mathbb{A}_c < b-V_c(\pi_k)$, and
> $\infty$ otherwise.
>
> **Why it doesn't approach TRPO:** If the extra constraint term $\beta\bar D_\phi$ were not present, we'd obtain the classical TRPO update. The reason why we don't get TRPO for $\beta\searrow 0$ is that the constraint term in the divergence doesn't vanish completely. Instead it approaches a step function, similar to how barrier functions behave in the interior point method.
>
> **Why it approaches CPO:** Constraining
>
> $$
> \bar D_{KL} +\mathbb{I}(\{\mathbb{A}_c < b-V_c(\pi)\}) < \delta
> $$
>
> is the same as constraining
>
> $$
> \bar D_{KL} < \delta
> $$
>
> and
>
> $$
> \mathbb{A}_c < b - V_c(\pi_k),
> $$
>
> which is what CPO does.
>
> > In Proposition 4, according to the proof in the appendix, , Why is the upper bound of C-TRPO smaller than that of CPO? Could the authors provide a more detailed explanation of this upper bound?
>
> We restate the bound here for reference:
> $$
> V_c(\pi_{k+1}) \leq V_c(\pi_{k}) + \Psi^{-1}(\delta/\beta) + \frac{\sqrt{2\delta} \gamma \epsilon_c}{1-\gamma}.
> $$
>
> It holds when $\pi_k$ is safe, hence we assume in the following that $V_c(\pi_{k}) \leq b$. The bound depends on the choice of hyperparameter $\beta$ through $\Psi^{-1}(\delta/\beta)$ for a fixed $\delta$.
>
> Specifically, $\Psi^{-1}$ is an increasing convex function of $\beta$. Its exact form depends on the safe mirror function term $\phi$. It atains its minimum for $\beta\to+\infty$, which is $0$, and maximum with $b-V_c(\pi_{k})$ for $\beta\searrow0$ (CPO).
>
> #### CPO bound:
>
> $$
> \beta \searrow 0 \Rightarrow V_c(\pi_{k+1}) < b + \frac{\sqrt{2\delta} \gamma \epsilon_c}{1-\gamma}
> $$
>
> #### Large $\beta$ bound:
>
> $$
> \beta \to +\infty \Rightarrow V_c(\pi_{k+1}) \leq V_c(\pi_{k}) + \frac{\sqrt{2\delta} \gamma \epsilon_c}{1-\gamma}
> $$
>
> This means that the bound will improve upon CPO as long as we choose a finite non-zero $\beta$.
>
> > As the policy approaches the constraint boundary, in equation 15 will approach infinity, which may make Equation (14) unsatisfiable and results in no solution. How is this situation addressed in the proposed framework?
>
> The answer depends on which policy is meant here. Let $\bar D_\phi(\pi||\pi_k)$ be defined as in (15) and let's call $\pi_k$ the target policy and $\pi$ the proposal policy. There are multiple cases that might be of interest:
>
> 1) The proposal policy oversteps: In the implementation we simply return a reserved symbol for infinity (e.g. numpy.inf in our case) if the policy advantage exceeds the threshold $b-V_c(\pi)$. This is ok because the divergence approaches infinity as the policy advantage approaches $b-V_c(\pi)$, and because, in theory, we could just as easily define the range of $\phi$ to be the extended reals $\mathbb{R}\cup\{+\infty\}$ (we update the definition of $\bar D_\phi$ to make this explicit).
> With a finite trust region radius $\delta \in \mathbb{R}$, the solution of equation (14) with a trust region divergence defined through (15) will have a solution as long as the "target" policy is safe. When the proposed gradient update oversteps, the divergence will equal $\infty$, but backtracking line search will still work as long as the implemented number system let's us check $\infty > \delta$. Backtracking line search can of course fail if we don't reserve enough backtracking steps, but this is true for TRPO, too.
>
> 2) The target policy oversteps: The divergence is indeed undefined, but also the requirements for equation 14 are not met, because $\pi_k$ is not in the safe set. In this case, we perform a recovery step (see algorithm 1).
>
> 3) The target policy approaches the constraint boundary: If $\pi_k$ get's really close to the constraint boundary, pre conditioning the policy gradient with the pseudo-inverse of the Hessian could lead to numerical issues. We did not encounter this problem in our experiments, but one could mitigate this issue by including a slack variable.
>
> ### Conclusion
>
> Thanks again for your feedback! We hope our responses address your comments. Please let us know if you have follow-up questions.

---

> > ### Comment · Reviewer_ujBB · 2024-11-27
> >
> > The reviewer thanks the authors for their responses. Unfortunately, their responses do not fully address my concerns.
> >
> > There are **inconsistencies** between the experimental results in the revision and those in the original manuscript, raising doubts about the **reliability of the results**. For example, in Figure 3 of the original manuscript, the reward of C-TRPO was lower than that of CPO, but in the revision, C-TRPO achieves the highest reward.
> > Moreover, the results are vague, undermining their reliability. For example, in Figure 3, the mean values across environments fail to reflect the algorithm's actual performance in each specific environment, and the same applies to Figure 10.
> >
> > Regarding weakness 2, which significantly impacts the algorithm’s solving process, their response does not adequately address my concerns.
> >
> > The authors' response also conflicts with the paper.
> > For example, (1) the response states $\bar D_C = \bar D_{KL} + \bar D_\phi$, which differs from the paper $\bar D_C = \bar D_{KL} + \beta \bar D_\phi$.
> > Based on Equations 9 and 10, $\beta$ is independent of $\bar D_\phi$.
> > (2) The revision claims that for a safe policy $\pi$, $V_C(\pi_k) \leq b$, not $V_C(\pi_k) < b$ as stated in the response. The response cannot prove that the bound for C-TRPO is strictly smaller than that of CPO.
> > Additionally, several new assumptions are introduced in the response that were not present in the original manuscript.
> >
> > Therefore, I think this paper still has many theoretical and experimental issues. Given these concerns, I remain doubtful about the validity of the proposed method and will maintain my score.

---

> ### Author Response · Authors · 2024-11-27
> **Reply to second response of Reviewer ujBB**
>
> Thank you for your response. We see that you still have some concerns, which we think can be addressed.
>
> **Validity of experiments**: We assume that you refer to the final return of C-TRPO and CPO in the bar chart in Figure 3 as opposed to the original sample efficiency curves. We re-ran the experiments for C-TRPO with $\beta=1$ (see response to your question about environment-wise experiments), and chose a bar chart to de-clutter Figure 3. The height of the bars represents the inter quartile mean, which is a sample statistic that naturally varies when we re-run the experiments. This is accounted for by the whiskers, which represent the bootstrap confidence intervals [2]. We don't claim that C-TRPO has higher return than CPO, and the figure should not be interpreted in this way. Hence, we kindly disagree with the claim that the results are not reliable.
>
> **Code:** Related to the question of the validity and reproducability of the experimental results, we would like to stress that we made our code avalable in an anonymous repo https://anonymous.4open.science/r/c-trpo/. It is entirely derived from the codebase of [1], which is available at https://github.com/PKU-Alignment/Safe-Policy-Optimization, where we only add the algorithms IPO, P3O, C-TRPO and CPO-Hysteresis, as well as data evaluations based on [2].
>
> **Minor typos in the response**: You are right, it should be $\bar D_C = \bar D_{KL} + \beta \bar D_\phi$, otherwise $\bar D_C$ does not depend on $\beta$ and our response would not make sense. You are also correct that in the response, safe policies should be $V_c(\pi_k)\leq b$ like in the manuscript, which has the correct definitions and assumptions. This is again just a typo, we edit our previous response to correct this. Further, the answer is still valid with the original assumption $V_c(\pi_k)\leq b$.
>
> **Validity of theoretical results**: Taking the typos in our response into account, the theoretical results are valid, no additional assumptions are needed. Mind that if $\Psi^{-1}(\beta/\delta)<b-V_c(\pi)$ (strictly less now), than the bound is strictly less than that of CPO. Due to the strict convexity of $\Psi$ and because it is **incresing** on the specified interval, $\Psi^{-1}(\beta/\delta)$ exists on the specified interval too, and is a strictly concave **increasing** function $\Psi^{-1}: [0, \infty) → [0, b-V_c(\pi))$, with maximum at $b - V_c(\pi_k)$, which is enough to prove that for any finite $\beta>0$, i.e. $\beta\in (0,\infty)$, it holds that $\Psi^{-1}(\beta/\delta)<b-V_c(\pi)$. To improve readability of the proof, we add a last sentense to explicitly states this.
>
> > The response cannot prove that the bound for C-TRPO is strictly smaller than that of CPO.
>
> In the response we only provide an explanation to help in understanding the result. The proof can be found in the appendix of the manuscript, where we add a sentense to make the proof more explicit.
>
> > Regarding weakness 2, which significantly impacts the algorithm’s solving process, their response does not adequately address my concerns.
>
> We want to stress that it is a core feature of the divergence to diverge to $+\infty$ at the constraint surface as this encourages policies to stay safe during optimization. Of course, this needs to be dealt with algorithmically. In our response to weakness 2 we answer with "see questions below", which refers to our answer to question 3, where we extensively discuss the ways in which the divergence could evaluate to $\infty$ and how our framework addresses this. Could you please clarify in which way our response to question 3 does not address your concerns and how we can improve it?
>
> > Therefore, I think this paper still has many theoretical and experimental issues. Given these concerns, I remain doubtful about the validity of the proposed method and will maintain my score.
>
> We hope that our answers adequately address your concerns and we are happy to answer any further questions.
>
> [1] Jiaming Ji, Borong Zhang, Jiayi Zhou, Xuehai Pan, Weidong Huang, Ruiyang Sun, Yiran Geng, Yifan Zhong, Josef Dai, and Yaodong Yang. Safety gymnasium: A unified safe reinforcement learning benchmark. In Thirty-seventh Conference on Neural Information Processing Systems Datasets and Benchmarks Track, 2023. URL https://openreview.net/forum?id=WZmlxIuIGR.
>
> [2] Rishabh Agarwal, Max Schwarzer, Pablo Samuel Castro, Aaron C Courville, and Marc Bellemare. Deep reinforcement learning at the edge of the statistical precipice. Advances in Neural Information Processing Systems, 34, 2021b.

---

### Official Review · Reviewer_hPGt · 2024-10-31

**Soundness:** 2
**Presentation:** 3
**Contribution:** 2
**Rating:** 5
**Confidence:** 2

**Summary:**

This paper introduces Constrained Trust Region Policy Optimization (C-TRPO), an approach that maintains safety constraints throughout reinforcement learning by shaping policy space trust regions to contain only safe policies. C-TRPO achieves competitive rewards and constraint satisfaction compared to leading CMDP algorithms, with theoretical convergence guarantees and experimental success in reducing constraint violations.

**Strengths:**

1. The idea of incorporating safety constraints into the trust region in TRPO is very reasonable and novel compared with penalty-based methods.
2. Both theoretical explanation of the C-TRPO and intuitive visualization in the toy MDP as in Figure 2 help to understand the effectiveness of C-TRPO, that it tries to behave safely in the trust region part instead of the target part.

**Weaknesses:**

1. One concern is the learning of the cost value V_c if this term is unknown. Since CPO suffers from estimation errors, C-TRPO has exactly the same problem. The theoretic analysis builds on the assumption that this function is accurate.
2. From the experimental results, the improvement over certain baselines is limited. For example, TRPO-Lag achieves smaller costs by the end of training and similar reward performance. Also in Table 1, CPO outperforms C-TRPO in many tasks.

**Questions:**

1. Is the action dimension two for the toy MDP used in Figure 2? Then the y-axis should represent a_2?
2. Line 167, D_k is not consistent with the previous Bregman divergence?

---

> ### Author Response · Authors · 2024-11-23
> **Response to Reviewer hPGt**
>
> # Weaknesses:
>
> > One concern is the learning of the cost value V_c if this term is unknown. Since CPO suffers from estimation errors, C-TRPO has exactly the same problem. The theoretic analysis builds on the assumption that this function is accurate.
>
> This is an issue for most safe policy optimization algorithms, including all tested baselines, and the newly added baselines. This is because in safe RL we don't assume to know the environment dynamics or the cost and reward functions. In most prior works, performance bounds are reported w.r.t. ideal value function estimates. Arguably, this is done 1) to provide insights into the proposed policy optimization problem in isolation from independant choices (like value function or advantage estimation), and 2) because, in principle, the estimation can be improved by increasing the number of steps sampled per update.
>
>
> > From the experimental results, the improvement over certain baselines is limited. For example, TRPO-Lag achieves smaller costs by the end of training and similar reward performance. Also in Table 1, CPO outperforms C-TRPO in many tasks.
>
> We would like to emphasise, that C-TRPO improves upon all baselines when we take into account the return *and* the cumulative constraint violation, which is not just about final cost but about constraint violation throughout training, which we think is a more suitable measure of safety during training. This problem has been studied before, see for example [1].
>
> [1] Yonathan Efroni, Shie Mannor, and Matteo Pirotta. Exploration-exploitation in constrained mdps,
> 2020. URL https://arxiv.org/abs/2003.02189
>
> # Questions:
>
> > Is the action dimension two for the toy MDP used in Figure 2? Then the y-axis should represent a_2?
>
> This is a mistake. It should be $a_1$ for all axes, but the conditioning state should change: $s_1$ on the x-axis and $s_2$ on the y-axis.
>
> > Line 167, D_k is not consistent with the previous Bregman divergence?
>
> We assume that the question arose since in the definition of the Bregman divergence $D_{\textup{K}}$ the policies explicitely appear. However, it can be shown that this Bregman divergence arises from the conditional entropy $\Phi_{\textup{K}}$ as we remark directly after the definition of $D_{\textup{K}}$ and provide a reference for this. We hope that this helps understanding the divergence $D_{\textup{K}}$. If we have misinterpreted your questions, please do not hesitate to let us know.
>
> # Conclusion
>
> Thanks again for helping us to improve our manuscript! We hope our responses address your comments. Please let us know if you have follow-up questions. We would be happy to discuss.

---

> > ### Comment · Reviewer_hPGt · 2024-12-02
> > **Response**
> >
> > Thank the authors for answering my questions. I think the imperfect critic makes a big difference to the paper's theoretical contribution, for which I am still not convinced by the current experiments. I will maintain my score.

---

> > > ### Author Response · Authors · 2024-12-02
> > >
> > > Thank you for your reply. Please note that it is standard to study the theoretical properties first without statistical estimation errors as sample complexity analysis in reinforcement can be delicate and technical, see for example the original works on TRPO and CPO. We see our main contribution in the proposition of a novel algorithm for CMDPs and we wanted to highlight that theoretical guarantees can be provided. Regarding the experimental side, we see our work as a proof of concept, where we provide a novel ansatz that naturally integrates the safety constraints in a way, which is much more natural from a convex optimization perspective. We provide an efficient implementation of this approach, which improves safety during training over existing baselines with no computational overhead and without compromise in performance. Please note that we have added multiple baselines and updated the presentation of our computational experiments. We do believe that this proof of concept and the fact that we improved over existing methods is a contribution that is worth communicating. Further, we think that it is hard to judge how much improvement over existing baselines is required to deserve publication.
> > >
> > > Therefore, we kindly ask you to reconsider, two points – sample complexity analysis and bigger improvement – are reasonable criteria for the decision of acceptance of a conceptual paper proposing a new methodology for CMPDs.

---

### Official Review · Reviewer_rTAs · 2024-11-03

**Soundness:** 3
**Presentation:** 3
**Contribution:** 3
**Rating:** 6
**Confidence:** 3

**Summary:**

The paper presents a novel method for safe RL, i.e. solving CMDPs while ensuring safety during training. The method is based on modifying trust region methods (i.e. TRPO and its constrained variant CPO) to yield trust regions that are contained in the set of safe policies. This is achieved by incorporating a barrier-like function to the trust region divergence, that approaches infinity as the expected cost of the updated policy approaches the threshold. The modified constrained objective is then approximately solved similarly to TRPO, with an additional recovery step in the case that an unfeasible point is reached. The authors provide a detailed theoretical analysis of their approach, and demonstrate the effectiveness of their method compared to other safe RL algorithms.

**Strengths:**

- The paper investigates an important problem and draws interesting connections to prior works on trust region methods.
- While the idea of using barrier functions for safe RL has been explored before in a number of works, the present paper provides an original and interesting theoretical connection based on modifying the Bregman divergence of trust region methods.
- The authors propose a simple yet effective recovery scheme for unfeasible policies.
- The main part of the paper is well-structured. Ideas are introduced clearly and it is explicitly shown how they relate to previous works.
- A further strength of the paper is the sound theoretical analysis of the proposed method, which is based on similar investigations for other trust region methods.
- The experimental results are promising: the method is shown to achieve competitive return with lower cost regret compared to the presented baselines.

**Weaknesses:**

While the paper provides worthwhile contributions, in my view there are several improvements that could be made. These mainly concern experimental results and exposition. If these concerns are accounted for, I am happy to increase my score.

Exposition
- The comparison to related work could be improved. In particular, while the related work section in the introduction summarises relevant approaches, it does not explicitly contrast them to the proposed method.
- For example, in the discussion of penalty methods it is stated that they introduce bias and produce suboptimal policies. Why does introducing a logarithmic barrier function to the Lagrangian (as in e.g. IPO) introduce more bias than modifying the trust region divergence?
- It would also be interesting to compare the theoretical bounds of the proposed method to those of other baselines besides CPO, e.g. IPO and the work by Ni et al. (2024).
- The discussion of relevant material in the background section focuses on the setting of discrete state and action spaces. However, one of the primary appeals of policy gradient methods is their applicability to the continuous setting (and this is indeed where the proposed method is evaluated). A discussion of how this relates to the introduced background would be appreciated.
- Furthermore, a brief discussion of Bregman divergences (possibly in the Appendix) would increase readability of the paper for readers not familiar with the topic.
- The experiments section is missing a (brief) discussion of the environments and associated constraints.

Experiments:
- The experimental evaluation does not include other approaches (e.g. P3O), particularly those also based on log-barriers (e.g. IPO, Ni et al. (2024)), which are relevant baselines.
- The ablation study on the hysteresis parameter shows that it is an important component of the achieved cost regret. The same idea can equally be applied to CPO. An ablation study comparing the proposed approach to CPO with hysteresis would highlight the effect of the main contribution of the paper, which is the modified trust region.

Minor remarks:
- The citation for IPO is wrong, this should be Liu et al. (2020) (line 71).
- $V_r^\pi(s)$ in Eq. 31 should be $V_{c_i}^\pi(s)$ (line 737).
- The definition of $L_\theta$ is missing in Eq. 40 (line 792).
- The presentation of Table 1 could be improved. Please highlight the best achieved cost in each row (e.g. bold or underline) and add standard deviations if possible. In the CarButton1 line, no return is bold.

**Questions:**

- How does the proposed approach compare to relevant baselines (e.g. IPO, Ni et al. (2024)), both in terms of theoretical bounds and empirical performance?
- Why does introducing a logarithmic barrier function to the Lagrangian (as in e.g. IPO) introduce more bias than modifying the trust region divergence?
- Can you provide an ablation study in which the proposed approach is compared to CPO with the same hysteresis scheme?

---

> ### Author Response · Authors · 2024-11-23
> **Response to Reviewer rTAs (pt 1/2)**
>
> Thank you for your time to review our submission, which we are convinced has helped us improve our manuscript. In the following, we want to address some of the points that you rightfully raised.
>
> # Weaknesses
>
> ## Exposition
>
> > The comparison to related work could be improved. In particular, while the related work section in the introduction summarises relevant approaches, it does not explicitly contrast them to the proposed method.
>
> We agree that our discussion of related works was rather short and have extended it significantly.
>
> > For example, in the discussion of penalty methods it is stated that they introduce bias and produce suboptimal policies. Why does introducing a logarithmic barrier function to the Lagrangian (as in e.g. IPO) introduce more bias than modifying the trust region divergence?
>
> See our answer below.
>
> > It would also be interesting to compare the theoretical bounds of the proposed method to those of other baselines besides CPO, e.g. IPO and the work by Ni et al. (2024).
>
> See our answer below.
>
> > The discussion of relevant material in the background section focuses on the setting of discrete state and action spaces. However, one of the primary appeals of policy gradient methods is their applicability to the continuous setting (and this is indeed where the proposed method is evaluated). A discussion of how this relates to the introduced background would be appreciated.
>
> In the presentation, we restrict ourselves to the finite setting for ease of presentation. But you are correct that it is important to also discuss the continuous setting. Indeed, in all of our experiments we encounter continuous state and action spaces. We have added a subsection (Appendix B.3) where we discuss the definition of C-TRPO in the continuous case.
>
> > Furthermore, a brief discussion of Bregman divergences (possibly in the Appendix) would increase readability of the paper for readers not familiar with the topic.
>
> We have followed your suggestion and have added a brief paragraph in Appendix A.
>
> > The experiments section is missing a (brief) discussion of the environments and associated constraints.
>
> ## Experiments
>
> > The experimental evaluation does not include other approaches (e.g. P3O), particularly those also based on log-barriers (e.g. IPO, Ni et al. (2024)), which are relevant baselines.
>
> We have now included four more baselines in our experiments, which are FOCOPS, CUP, IPO and P3O. Unfortunately, Ni et al. (2024) did not provide implementation details. Still, methods which have similar or smaller cost regret have much smaller return. These results are in line with our previous findings and support our conclusions.
>
> > The ablation study on the hysteresis parameter shows that it is an important component of the achieved cost regret. The same idea can equally be applied to CPO. An ablation study comparing the proposed approach to CPO with hysteresis would highlight the effect of the main contribution of the paper, which is the modified trust region.
>
> See our reply below.
>
> > Minor remarks: ...
>
> We updated the pdf to incorporate the suggestions.
>
> > The presentation of Table 1 could be improved. Please highlight the best achieved cost in each row (e.g. bold or underline) and add standard deviations if possible. In the CarButton1 line, no return is bold.
>
> Thank you for the suggestion, see the updated table in the new pdf.
>
> We would like to emphasize that the information this table provides is limited, because it doesn't show the cost violations during training, which is our main motivation for developing C-TRPO. As we mentioned in the main text, we only provide it "for completeness". For a full picture, it is best to compare the methods based on Figure 3.

---

> > ### Author Response · Authors · 2024-11-23
> > **Response to Reviewer rTAs (pt 2/2)**
> >
> > # Questions
> >
> > > How does the proposed approach compare to relevant baselines (e.g. IPO, Ni et al. (2024)), both in terms of theoretical bounds and empirical performance?
> >
> > Ni et al. (2024) theoretically study the convergence of a first-order stochastic gradient method with a logarithmic barrier. They obtain a result that shows safety during training with high probability and guarantees exponential decay of the suboptimality gap up to a statistical and function approximation error, where similar results have been obtained for a variety of (natural) policy gradient methods, see for example Agarwal et al. (2021) and follow up works. We believe that a similar strategy can be pursuit for C-TRPO and C-NPG. However, due to the technicality of the subject and the page limit policy, we leave a concise convergence analysis for future works.
> >
> > Unfortunately, Ni et al. (2024) do not provide implementation details, hence, we can not directly incorporate LB-SGD as a baseline in our experiments. However, we add IPO, P3O, FOCOPS, and CUP to the experiments, see general reply.
> >
> > > Why does introducing a logarithmic barrier function to the Lagrangian (as in e.g. IPO) introduce more bias than modifying the trust region divergence?
> >
> > Adding a logarithmic barrier function or a different penalty like an entropic regularization changes the objective function. In particular, this changes the set of optimizers and introduces an additional error, commonly referred to as the regularization error, which has to be estimated. Note that changing the trust regions and therefore the problem geometry does not change the objective function and hence the set of optimizers. Therefore, in contrast to penalty methods a change of the trust region does not introduce an additional error.
> >
> > > Can you provide an ablation study in which the proposed approach is compared to CPO with the same hysteresis scheme?
> >
> > Thank you for this excellent question, we agree that this makes a lot of sense here. We have conducted an ablation study comparing the performance of both CPO and C-TRPO both with and without hysteresis. For both algorithms, the hysteresis reduces the cost regret, however, already C-TRPO without hysteresis achieves smaller cost regret compared to CPO with hysteresis. In conclusion, we think that the ablation study is important to include and strengthens our submission.

---

### Author Response · Authors · 2024-11-23
**General Reply**

We want to thank all reviewers for their thorough evaluation of our submission and provide extensive answers to the individual points raised by the reviewers below.

Here, we want to elaborate on the experimental evaluation as well as our motivation and relation to existing works as these points were raised by multiple reviewers. We add the following discussion to the main text (changes in blue).

1. **Experiments:** We have improved our experimental evaluation of C-TRPO in the following ways:
+ *Ablation study of hysteresis:* We agree with the criticism. We have conducted an ablation study comparing the performance of both CPO and C-TRPO both with and without hysteresis. For both algorithms, the hysteresis reduces the cost regret, however, already C-TRPO without hysteresis achieves smaller cost regret compared to CPO with hysteresis. You find the new plots in the PDF.
+ *Recent baselines:* We further include 2 recent trust region methods (FOCOPS and CUP) and 2 penalty methods (IPO and P3O) in our experimental evaluation and discussions. We leave LB-SGD out of the experimental evaluation, because Ni et al. 2024 don't provide implementation details. Because Figure 3 got too cluttered, we change it to a bar chart showing only the last iterates.


2. **Bias of penalty methods**: There were multiple questions regarding the bias introduced by penalty methods. Adding a penalty function like a logarithmic barrier changes the objective function. In particular, this changes the set of optimizers and introduces an additional error, commonly referred to as the regularization error, which has to be controlled. Note that changing the trust regions and therefore the problem geometry does not change the objective function and hence the set of optimizers. Therefore, in contrast to penalty methods a change of the trust region does not introduce an additional error. We have added a short explanation of this in the introduction. Please let us know if there are more questions regarding the bias introduced by penalization.

3. **Exposition, motivation, relation to existing works, and novelty:** We have tried our best to outline our idea and motivation for the design of safe trust regions and our intuition why they are useful. In particular, we have changed the following things:

    We have added the following discussions:
    + We have expanded the related works paragraph and add a motivational paragraph after that in the introduction.
    + We have expanded the beginning of section 3 for improved exposition of the technical details.

    Here, we briefly summarize the main points of our argumentation.
    + *Difference to CPO:* Current approaches only incorporate the safety constraints via the objective function. We incorporate them into the geometry of the policy space. This can be seen as an extension of TRPO to the constrained case, which is conceptually very different to CPO. The updates of CPO only differ from those of TRPO when TRPO would leave the safe region and  allows for updates that lie at the constraint surface. In comparison, C-TRPO uses trust regions that are flatter parallel to the constrained surface. This naturally encourages updates to move more parallel to the constraint surfaces and thereby reduces the likelihood of constraint violations during training compared to CPO. We see the following advantages.
    + *Properties of C-TRPO:* The following properties of C-TRPO are beneficial compared to existing methods:
        + C-TRPO significantly improves constraint satisfaction of trust region methods while achieving competitive returns.
        + As C-TRPO is a true trust region method, there is no need for projection or constrained minization in an inner loop, which simplifies computations.
        + Incorporating constraints into the trust region yields a true trust region method and feels natural from a convex optimization point of view, see for example Alvarez et al. (2004).
        + We provide a natural policy gradient scheme corresponding to C-TRPO, which is missing for other methods like CPO.

    Further, we have streamlined our notation and presentation of the theoretical results to improve readability of the theoretical sections.

Thanks again for your helpful comments and questions. We are convinced that they have helped us to substantially strengthen the manuscript and hope we addressed them to your satisfaction.

We remain attentive to your feedback!

---

### Meta-Review · Area_Chair_krrU · 2024-12-19

**Metareview:**

Embedding Safety into RL: A New Take on Trust Region Methods

Summary: The paper introduces a new approach for safe reinforcement learning (RL) using constrained trust region policy optimization (C-TRPO). The method modifies traditional trust region policy optimization (TRPO) by altering the geometry of policy updates to ensure safety constraints are respected throughout training. By embedding safety directly into the policy update mechanism, C-TRPO creates trust regions consisting exclusively of safe policies without requiring explicit projections or constrained optimization in the inner loop. The paper also proposes a natural policy gradient variant (C-NPG) for continuous-time optimization, offering theoretical guarantees for safety and convergence to optimal policies. Empirical evaluations on Safety Gym benchmarks show that C-TRPO achieves competitive returns while reducing constraint violations compared to state-of-the-art methods.


Comments: We received five expert reviews, with the scores  3, 5, 5, 6, 8, and the average score is 5.40.

Reviewers have given positive comments about multiple aspects of the paper. This basic idea is an interesting extension of the TRPO to the constrained settings. The paper is well written and the theoretical analysis is rigorous.  The hysteresis-based recovery mechanism is also a useful addition to improve constraint satisfaction during training.

However, many reviewers have pointed out a number of weaknesses in the paper. Reviewer rTAs has multiple comments about missing comparisons, both in terms of literature review and in terms of experiments,  with relevant works. Reviewer ujBB has pointed out a few concerns about the theoretical analysis and experimental results, and some apparent inconsistency between them. While the authors have provided a detailed response, the reviewer is still not completely convinced about the improvements. Reviewer C9Zt. More than one reviewer has expressed concerns about novelty, especially compared to the CPO algorithm.  Reviewers have also given detailed comments about improving the quality of the presentation.

Overall, the paper demonstrates a promising approach to embedding safety into trust-region methods and contributes to the field of safe RL. However, the paper does not convey the novelty of the proposed approach effectively and is missing comparisons with many relevant works. Addressing these concerns could significantly improve the paper's quality and relevance. I encourage the authors to revise and resubmit after addressing the outlined weaknesses.

**Additional Comments On Reviewer Discussion:**

Please see the "Comments" in the meta-review.

---

### Decision · Program_Chairs · 2025-01-22

Reject